# Differences in eye movements for face recognition between Canadian and Chinese participants are not modulated by social orientation

**Francis Gingras**[1,2‡], **Amanda Estéphan**[1,2‡], **Daniel Fiset**[1], **He Lingnan**[3], **Roberto Caldara**[4]*, **Caroline Blais**[1]*

1 Département de psychoéducation et psychologie, Université du Québec en Outaouais, Gatineau, Canada, 2 Département de psychologie, Université du Québec à Montréal, Montreal, Canada, 3 School of Communication and Design, Sun Yat-Sen University, Guangzhou, People's Republic of China, 4 Eye and Brain Mapping Laboratory (iBMLab), Department of Psychology, University of Fribourg, Fribourg, Switzerland

‡ FG and AE are co-first authors on this work.
* caroline.blais@uqo.ca (CB); roberto.caldara@unifr.ch (RC)

**Data Availability Statement:** Data is available at the following OSF location: https://osf.io/b5tdy/

**Funding:** This work was supported by grants from the Natural Sciences and Engineering Research

## Abstract

Face recognition strategies do not generalize across individuals. Many studies have reported robust cultural differences between West Europeans/North Americans and East Asians in eye movement strategies during face recognition. The social orientation hypothesis posits that individualistic vs. collectivistic (IND/COL) value systems, respectively defining West European/North American and East Asian societies, would be at the root of many cultural differences in visual perception. Whether social orientation is also responsible for such cultural contrast in face recognition remains to be clarified. To this aim, we conducted two experiments with West European/North American and Chinese observers. In Experiment 1, we probed the existence of a link between IND/COL social values and eye movements during face recognition, by using an IND/COL priming paradigm. In Experiment 2, we dissected the latter relationship in greater depth, by using two IND/COL questionnaires, including sub-dimensions to those concepts. In both studies, cultural differences in fixation patterns were revealed between West European/North American and East Asian observers. Priming IND/COL values did not modulate eye movement visual sampling strategies, and only specific subdimensions of the IND/COL questionnaires were associated with distinct eye-movement patterns. Altogether, we show that the typical contrast between IND/COL cannot fully account cultural differences in eye movement strategies for face recognition. Cultural differences in eye movements for faces might originate from mechanisms distinct from social orientation.

Council of Canada (NSERC; https://www.nserc-crsng.gc.ca/) to C.B. (RGPIN-2019-06201) and D. F. (RGPIN-2012-402513), and by a Canada Research Chair in Cognitive and Social Vision (# 950-232282) held by C.B. (https://www.chairs-chaires.gc.ca/chairholders-titulaires/profile-eng.aspx?profileId=4770#:~:text=Caroline%20Blais%2C%20Canada%20Research%20Chair,aims%20to%20increase%20this%20understanding.) The funders had no role in study design, data collection and analysis, decision to publish, or preparation of the manuscript.

**Competing interests:** The authors have declared that no competing interests exist.

## Introduction

Perception is shaped by experience and adaptation to the environment [1]. Our visual and social surroundings are molded by distinct cultural norms and values. Culture can be viewed as a vehicle of tradition of thought through which interpretation of the world is formed. During the last two decades, growing evidence arising from different fields has shown that interiorization of cultural norms plays a role in shaping perception [2, 3]. These findings have demonstrated that mechanisms of perception are not universal, as they are intrinsically linked to cultural backgrounds and social structures specific to geographical locations. This constructivist way of envisioning perception has given rise to many cross-cultural studies in this field, spanning from optical illusions [4] and color perception [5] to facial expression recognition [6]. Culture might determine the way we structure ourselves, how we interact with the world and how we perceive the environment.

It has been demonstrated that face recognition strategies do not generalize across individuals [7–10] and are reliable over time [11]. Notably, studies have revealed robust differences between individuals from East Asian and West European/North American countries ranging from eye movement strategies and information sampling resolution (i.e. spatial frequencies) during face recognition to emotion perception [3, 6, 12]. Specifically, for face recognition, evidence suggests East Asians adopt a more global attentional strategy involving more central fixations [13]. This has been related to peripheral processing of facial features [14, 15] and lower spatial frequency use in East Asian populations [10, 16, 17]. On the other hand, West European and North American individuals adopt a more local attentional strategy, involving fixations towards facial features and higher spatial frequency sampling. These results are important, as face-to-face contacts play a central role in social interactions. Nevertheless, exactly what cultural factors are responsible for such perceptual differences remains to be understood.

The social orientation hypothesis posits individualistic vs. collectivistic (IND/COL) value systems [18–21] as a major factor of explanation for many cultural differences in visual perception [2, 22] (however, see [23, 24]). This framework also includes the concept of independent and interdependent self-construals, as studies have shown that independent self-construal correlates with individualism and interdependent self-construal with collectivism [25]. This theory suggests that collectivistic values, more prevalent in East-Asians, would be linked to a holistic cognitive style, while individualistic values, more common in North Americans/West Europeans, would be linked to an analytic cognitive style. In terms of visual perception, this would translate into broader and narrower visual attention, respectively. This view is supported by evidence of differences between East Asians and West Europeans/North Americans along the dimensions of Individualism/Collectivism and Independence/Interdependence [19, 25], that are associated with differences in visual attention patterns [22].

For instance, when asked to judge the orientation of a line placed inside an independently rotating frame, Chinese participants are more influenced by the orientation of the frame than American participants [26, 27]. Later studies highlighted that Japanese participants were also more influenced by a changing background when asked to memorize foreground objects in a scene [28, 29]. The latter differences in scene memorization between Japanese and American participants were further supported by divergent eye movement patterns. In that study, Chinese participants gazed at the background for a longer period and started directing fixations to foreground objects later than American participants [30–33].

However, there have also been studies showing evidence that the social orientation hypothesis may not be what drives the observed differences between groups. Knox & Wolohan [23] have compared Chinese individuals raised in England to both English and Chinese participants, and have shown that while they are closer to British participant in their values, their eye

movement patterns are more similar to those of the Chinese group. Another study by Kardan et al. [24] comparing collectivist Mayan to US participants has shown that participants from the US fixate *more* on the background information during videos compared to Mayan children, which is not what would be expected with the social orientation hypothesis. While the authors do advise caution in applying the social orientation theory to their findings (it is unclear whether the individualism-collectivism continuum applies to Mayan culture), it still shows that not all of the influences attributed to culture are as clear-cut as they were expected to be.

Some studies have sought to more directly link social orientation to visual processing. Namely, a causal link between social orientation and perception was demonstrated using independent and interdependent priming [34]. The study results showed that "independence primed" participants were quicker to identify the small letters compared to the large letters in Navon figures, and the opposite was true for "interdependent primed" participants. Another study by Chua et al. [35] as shown that independent self-construal predicts worse performance in holistic perceptual learning. These studies suggest that social orientation plays a critical role in shaping cultural perceptual differences between West European/North American and East Asian observers, whereby collectivism/interdependence is associated with global attention and individualism/independence is associated with local attention.

Despite the aforementioned evidence, whether social orientation can also explain the cultural differences in face recognition remains unclear. A recent study using social self-construal priming on a Chinese population found that some instances of face processing, such as holistic processing and race categorization, were indeed affected by social self-construal priming, but that face recognition was not [36]. However, this study only included a sample of Chinese participants, limiting the generalizability of its findings and their genuine attribution to cross-cultural causes. Another study, with a sample of British-born Chinese participants, has highlighted a possible link between individual social values and facial recognition strategies. In their research paper, Kelly et al. [37] report a non-significant but considerable relationship between individualism/ collectivism and eye movement patterns during face recognition. Individualistic individuals were more likely to exhibit a fixation pattern typically associated with West Europeans and North Americans, whereas collectivistic individuals were more likely to exhibit a fixation pattern typically associated with East Asians. However, the sample size was quite small in those groups, likely not enough for sufficient statistical power, and only one questionnaire was used to measure individualism/collectivism.

To clarify these important issues, we investigated whether social orientation could account for cultural differences in eye movements during face recognition. We conducted two experiments to test if both primed and self-reported social orientation modulate eye movements during an Old/New face recognition task [13, 38]. To be consistent with previous social orientation studies, we have chosen designs that are similar to previous studies, both in the methods and questionnaires used. In Experiment 1, we probed the existence of a link between individualism/collectivism (IND/COL) and eye movements during face recognition by implementing an IND/COL priming paradigm. We used a within-subjects design as they have been shown to have better statistical power in priming paradigms [39]. This method is similar to the one used in Liu et al. [36]. However, unlike the latter study, we included both West European and Chinese participants.

In Experiment 2, we investigated the relationship between social orientation and facial recognition in greater depth by using two IND/COL questionnaires and including subdimensions to those concepts. We used the validated Auckland Individualism and Collectivism Scale (AICS), developed by Shulruf et al. [40], as well as a validated shortened version of the Horizontal and Vertical Individualism and Collectivism Scale (HVICS), originally developed by

Triandis & Gelfand [41]. Both questionnaires were validated in the target populations [42, 43] This method is akin to the protocol used in Kelly et al. [38]. Once again, we included participants from both cultures (North America and China).

Culture is a broad term that refers to a wide variety of concepts (ex. social norm, upbringing, belief systems, geographic location, etc.). Previous research has shown that country borders represent a relatively good proxy of cultural variations [44, 45]. In line with this idea, in the present article we define groups of individuals coming from East-Asian and Western countries as "culturally different", in the sense that they have grown up in environments with different sets of values, social norms, and so on.

## Experiment 1

### General information

West European and North American participants had little to no experience with East Asian cultures; Chinese participants had little to no experience with West European and North American cultures. To assess participants' experience with other cultures, we asked them to indicate if they had ever lived in a country other than their birth country. All participants were aged between eighteen and thirty-five years old and had normal or corrected-to-normal vision. Participants were recruited between 2015 and 2018. Task instructions, priming texts, and questionnaires were given to participants in their native language. Experimental protocols used in this study were approved by the Institutional Review Boards of University of Fribourg (Switzerland), Université du Québec en Outaouais (Canada), and Sun Yat-Sen University (China). Participants gave their written, informed consent to participate in the study and signed the consent form prior to the experiment. The entire study was conducted in conformity with the Code of Ethics of the World Medical Association (Declaration of Helsinki). Participant data was tied to an anonymous ID after data collection. Data analyses were conducted without any information, other than this ID, that could be linked to a participant's identity.

### Hypothesis

In line with Liu et al. [36], we hypothesized that group differences based on social orientation should mirror the differences between East Asians and West Europeans typically observed in face recognition strategies. Observers primed with a collective self-construal should exhibit more central fixations (i.e., nose area) whereas observers primed with an independent self-construal should exhibit more featural fixations (i.e., eyes and mouth areas).

### Participants

Fifteen Swiss individuals and twenty-one Chinese individuals took part in the experiment. Swiss participants were tested in Fribourg (Switzerland) and were born and lived their entire life in Switzerland. Chinese participants were tested in Guangzhou (China) and were born and lived their entire life in China. Our sample size was selected according to a G*Power assessment for a Repeated measures ANOVA with a within-between interaction, given a medium effect size (f = .25), a minimal statistical power of 80%, $\alpha$ = .05, nb groups = 2 and nb measurements = 2. Other parameters were kept at the default values. A minimum sample size of 34 participants was recommended from these settings. This power assessment was made for the cultural group x priming condition mixed ANOVA and the effect size was based on previous studies that found large Cohen's d effect sizes [13, 46]. Here, we used a smaller effect size than what was found in those studies with respect to cultural differences in eye movements during face recognition.

## Material and stimuli

Images were 382×390 pixels in size and displayed with a gray screen-wide background on an 800×600 pixel Dell P11302100 (refresh rate of 170 Hz). Participants were seated at a viewing distance of 70 cm from the computer screen, stimuli subtending 15 degrees of visual angle horizontally.

The face dataset used in this study is the one used in Blais et al. [13] There were a total of 280 identities with an equal number of males and females. Each identity featured three facial expressions (happy, disgusted and neutral). The decision to include different facial expressions replicates Blais et al. [13] and limits the use of "template-matching" to identify faces. Participants were explicitly asked to focus on memorizing the identity and then recognize the face regardless of the displayed facial expression. White European face images were drawn from the KDEF database [47], and East Asian face images were drawn from the AFID database [48]. Faces from both databases faced forward. Accidental local features such as brown spots, rashes and facial hair were removed using the Photoshop software. Faces had no distinctive features such as glasses, jewelry, scarf, etc. All faces were aligned as well as possible, using as parameter the least-square measure, on the positions of eyes, nose and mouth–by means of translation, rotation, and scaling. Face images were all gray scale and normalized in terms of luminance. In order to maintain some degree of ecological validity, faces were otherwise presented with their normal hair and facial contour.

## Eye-tracking apparatus

Eye movements were recorded at a sampling rate of 1000 Hz with the SR Research Desktop-Mount EyeLink 2K eye-tracker (average gaze position error of .25 degrees; spatial resolution of .01 degrees); only the dominant eye was tracked. In both experiments, participants' dominant eye was determined using a variation of the Miles test [49], similar to the "hole in the hand test [50]". To ensure that viewing distance was constant throughout the entire experiment and that eyes were accurately tracked, participants were asked to position their head on a chin and forehead rest, facing the screen at the appropriate viewing distance.

A nine-point manual calibration of ocular fixations was performed before each task block (see protocol section for details regarding "blocks"). Calibrations were then validated with the EyeLink software and repeated when necessary until the optimal calibration criterion was reached. At the beginning of each trial, participants were instructed to fixate a dot at the center of the screen to perform a drift correction. The experiment was implemented in MATLAB R2012b, using the Psychophysics [51–53] and EyeLink [54] toolboxes (see http://psychtoolbox.org).

## Protocol

An Old/New face recognition task along with an "Individualist (I) / Collectivist (We)" pronoun circling priming task [55] was used for this experiment. Overall, the experimental procedure went as follows.

Participants had to complete four "learning / recognition" blocks. Before each block, participants had to read one of four priming texts that were either individualist-oriented (written with first person singular pronouns, e.g. "I") or collectivist-oriented (written with first person plural pronouns, e.g. "We"). Ultimately, all participants read four texts, two individualist-oriented (once before learning White European faces; once before learning Chinese faces) and two collectivist-oriented (once before learning White European faces; once before learning Chinese faces), and thus all were subjected to both priming conditions (i.e., "I" or "We"-oriented texts). Half of the participants were primed with the two "I" texts first, the other half

with the two "We" texts first. The order of face ethnicities was counterbalanced across participants. Participants were asked to carefully read each priming text. As a means of priming, they were instructed to circle all pronouns included in the text. To make sure participants paid close attention to the textual content, they were informed that they had to answer questions about the story once they finished.

During learning periods, participants were instructed to memorize a series of fourteen different faces, presented to them one at a time for five seconds. During recognition periods, participants were presented with a second series of faces, half of which were previously memorized during the corresponding learning period, and were instructed to use the keyboard to indicate whether each face was one they had seen before (old; 'A' button on keyboard) or not (new; 'L' button on keyboard). Once again, faces were displayed one by one, but this time appeared on screen until a response. Faces appeared in the corners of the screen, so that first fixations were voluntarily aimed at a specific point in the face, instead of being centered on the initial fixation point. See Fig 1 for an illustration of the Old/New experimental design.

## Analyses

Fixation maps were produced for each participant and each experimental condition (i.e., priming situation). Fixation maps were computed by using fixation durations along image-centered coordinates. Fixation durations across individual maps were transformed into z-scores using the map fixation duration mean and standard deviation. For both experiments 1 and 2, we chose to analyze all eye movements produced during learning (5 seconds) and recognition (until response), again following the procedure of Blais et al. [13].

Learning and recognition periods were always analyzed separately. Eye movements were first analyzed using the iMap4 Linear Mixed Model [56] (LMM). To obtain the LMM parameters, iMap4 performs a pixel-by-pixel comparison of the fixation maps across participants and conditions. Then, a bootstrap procedure is applied to correct for multiple comparisons (i.e., across the pixels of the fixation maps). A fixed effect mixed ANOVA with "culture" as between factor and "priming" as within factor was performed on the resulting maps (alpha = .05; bootstraps = 1000). "Face ethnicity" was not considered since no effect of this variable was found in the original Blais et al. [13] study, as well as many subsequent replication studies [10, 14, 16, 38] (however, see [57, 58]).

As a second step, we investigated the effect of culture and priming on specific face regions where significant differences were previously found between West Europeans and East Asians. To this effect, we conducted a Region Of Interest (ROI) analysis on the corresponding eyes, nose, and mouth regions of the fixation maps; a mean of the face stimuli used for the experiment was taken as a template to delimit the ROIs (see Fig 2). These regions were selected as they are important in distinguishing respectively West European fixation patterns (feature-oriented) and East Asian fixation patterns (center-oriented). We also included in our analysis a ROI corresponding to the rest of the face, around the eyes, nose, mouth, as was done in Blais et al. [13]. Percentage of fixations allocated to the eyes, nose, mouth, and rest of the face respectively were compared between both cultural groups (Swiss and Chinese), and between both priming conditions ("I" and "We" personal pronoun priming). To examine group differences on fixation percentages for each ROI, we performed "Culture" x "Priming" ANOVAs on each ROI separately. The bilateral significance threshold (p < .025) was corrected for multiple comparisons using a Bonferroni correction, where differences on each ROI are tested at p = .025/4 = .006.

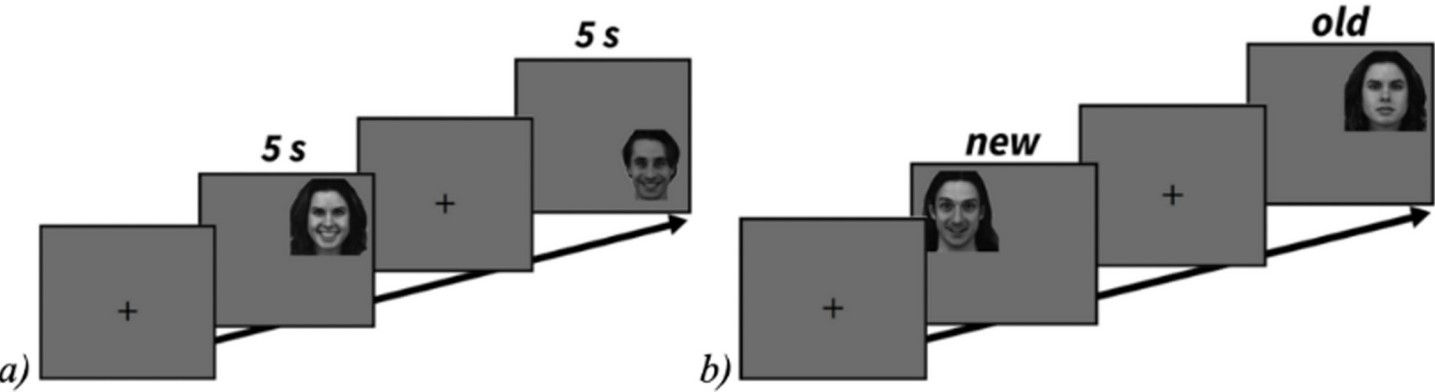

**Fig 1. Old/New experimental design.** Old/New face recognition experimental design and examples of stimuli: a) learning period (sequence of 14 faces per block; presentation time of 5 seconds per face); b) recognition period (sequence of 28 faces -- 14 new -- per block; presentation until response).

## Results

### Culture, social orientation priming, and eye movements

Fixation maps (see Fig 3) show that there is a trend toward the typically observed cultural differences in fixation patterns (more fixations on eyes for West Europeans; more fixations on center for East Asians), especially for learning periods. As for "priming conditions", on the other hand, no trend is observed (see Fig 4).

The mixed ANOVA (culture x priming) analysis with iMap4 yielded no significant results. This means that pixel-by-pixel differences in fixation maps for the main effect of culture, the main effect of priming and the culture x priming interaction were not revealed at least 95% of the time. In addition, mixed ANOVAs yielded no main effect of priming on any ROI, and no significant interactions between culture and priming. See Table 1 for detailed ROI-related statistics.

Interestingly, there was only an effect of culture on fixations over the rest of the face which was marginal for learning [$F(1,34)$ = 7.590; $p$ = .009; $\eta2p$ = .182] and significant for recognition [$F(1,34)$ = 9.230; $p$ = .005; $\eta2p$ = .213] periods. Independent Samples t-tests confirmed that Chinese participants distributed their fixations in the "rest of face" ROI more than Swiss participants during learning [$t(34)$ = -2.750; $p$ = .009; Cohen's d = -.931] and recognition [$t(34)$ = -3.040; $p$ = .005; Cohen's d = -1.030] periods. Average fixation percentage differences between cultures and between priming conditions are reported in Table 1 for each ROI.

## Discussion

In experiment 1, we revealed significant cultural differences that were atypical compared to previous results, and priming did not seem to have any effect on participant. It is unclear whether this is because there is no effect of priming or because the priming procedure simply failed. Our data show a qualitative cultural contrast, with more fixations towards the eyes for West Europeans and central fixations for East Asians. Analyses with iMap4 yielded no significant results. However, subsequent ROI analyses did reveal greater fixation distribution around the face for Chinese participants compared to Swiss participants. This supports the idea that Chinese observers spend more time than Swiss observers fixating areas peripheral to the main features. However, it differs from previous studies in that Chinese and Swiss participants did not differ in the time spent fixating the main facial features nor the center of the face. This difference may be due to the fact that our Chinese sample was recruited from a different region

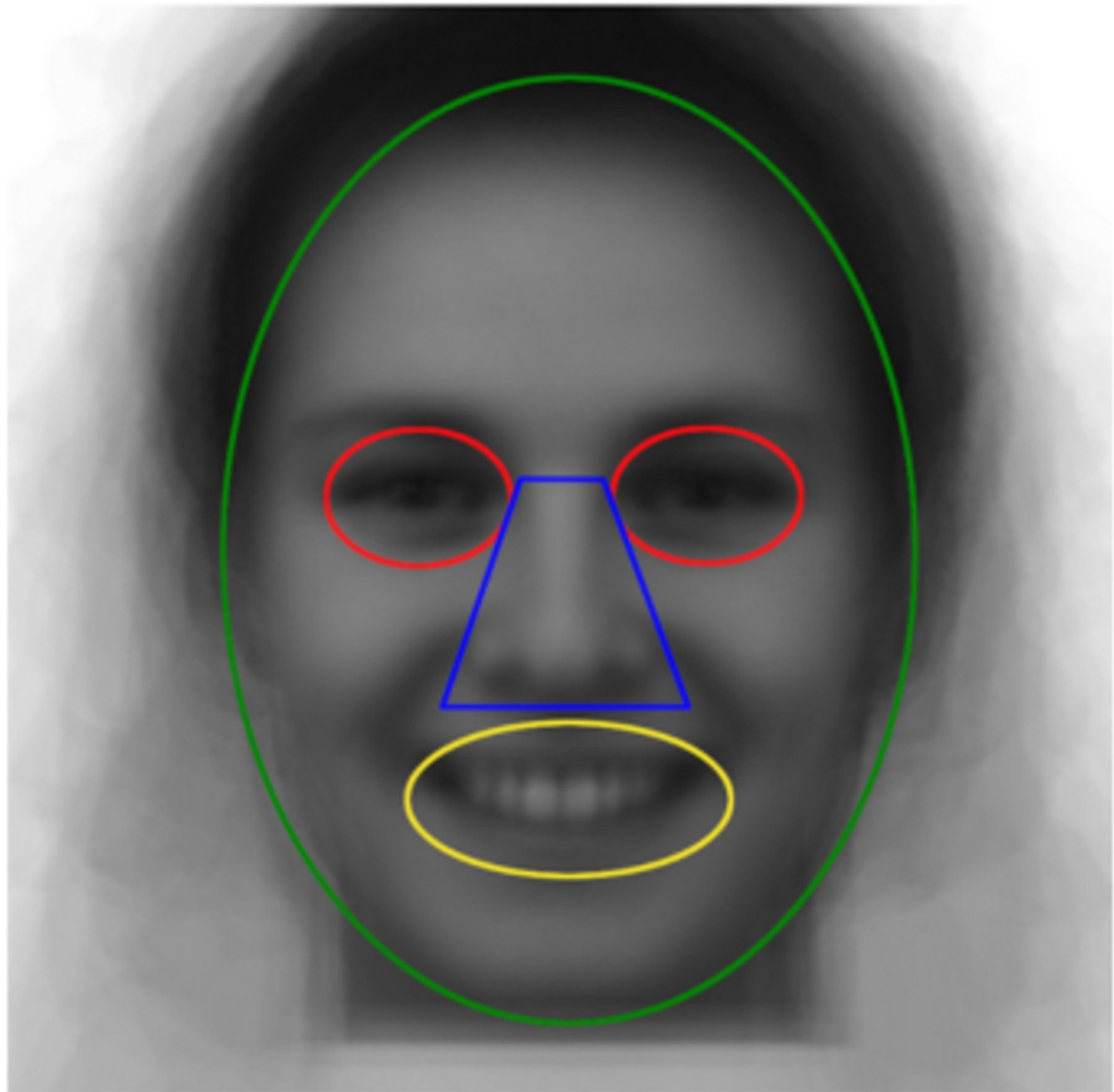

**Fig 2. Face ROI template.** Note. Average face stimulus with ROI corresponding to eyes (red), nose (blue), mouth (yellow), and rest of face (green).

than the one from Blais et al.[13]. This is still consistent with the idea proposed by Caldara et al. [14], Miellet et al. [15], Tardif et al. [10] and Estéphan et al. [16] that East-Asians process facial features in peripheral vision. Further investigation would be necessary to examine how samples collected from different regions within the same country may show different patterns of visual strategies.

Since iMap4 is especially conservative compared to the versions used in previous studies, the General Linear Model might have downplayed the effect of culture considering the

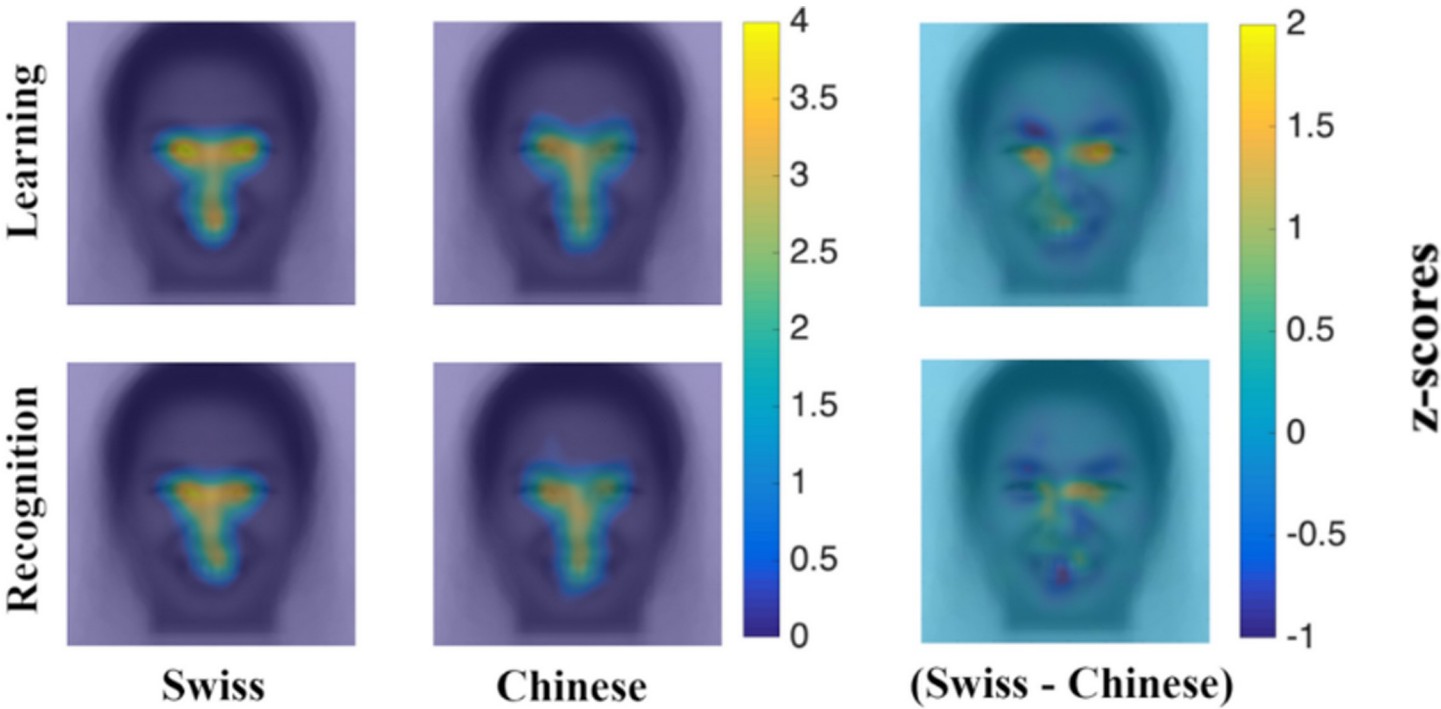

**Fig 3. Fixation maps–cultural differences.** Note. First column illustrates Swiss participants' average fixation maps for learning and recognition periods respectively; second column illustrates Chinese participants' average fixation maps for learning and recognition periods respectively; third column illustrates Swiss (yellow) and Chinese (blue) fixation biases for learning and recognition periods respectively (no significant cultural differences were revealed by iMap4). The term "fixation bias" refers to fixation locations that are associated more with one group compared to the other.

presence of other variables in the model with the present data. Moreover, the absence of robust significant differences between our cultural groups (with iMap4) might be related to individual differences in our sample, which could have been amplified by the "priming conditions" causing partial, but non-significant, interference. This observation emphasizes the importance of further investigating how individual social values could be associated with facial recognition strategies, which will be central to Experiment 2.

## Experiment 2

### Hypothesis

In the context of this experiment, we hypothesized that if social orientation differences between East Asians and North Americans explain the observed differences in face recognition strategies between those two groups, individuals with higher scores on the collectivism scales should exhibit more central fixations (i.e., the nose area), whereas individuals with higher scores on the individualism scales should exhibit more featural fixations (i.e., the eye and mouth areas).

### Participants

Sixty-three Canadian individuals and forty-nine Chinese individuals took part in the experiment. Canadian participants were tested in Gatineau (Canada), and were born and lived their entire life in Canada. Chinese participants were tested in Guangzhou (China) and were born and lived their entire life in China. All one hundred and twelve participants completed the

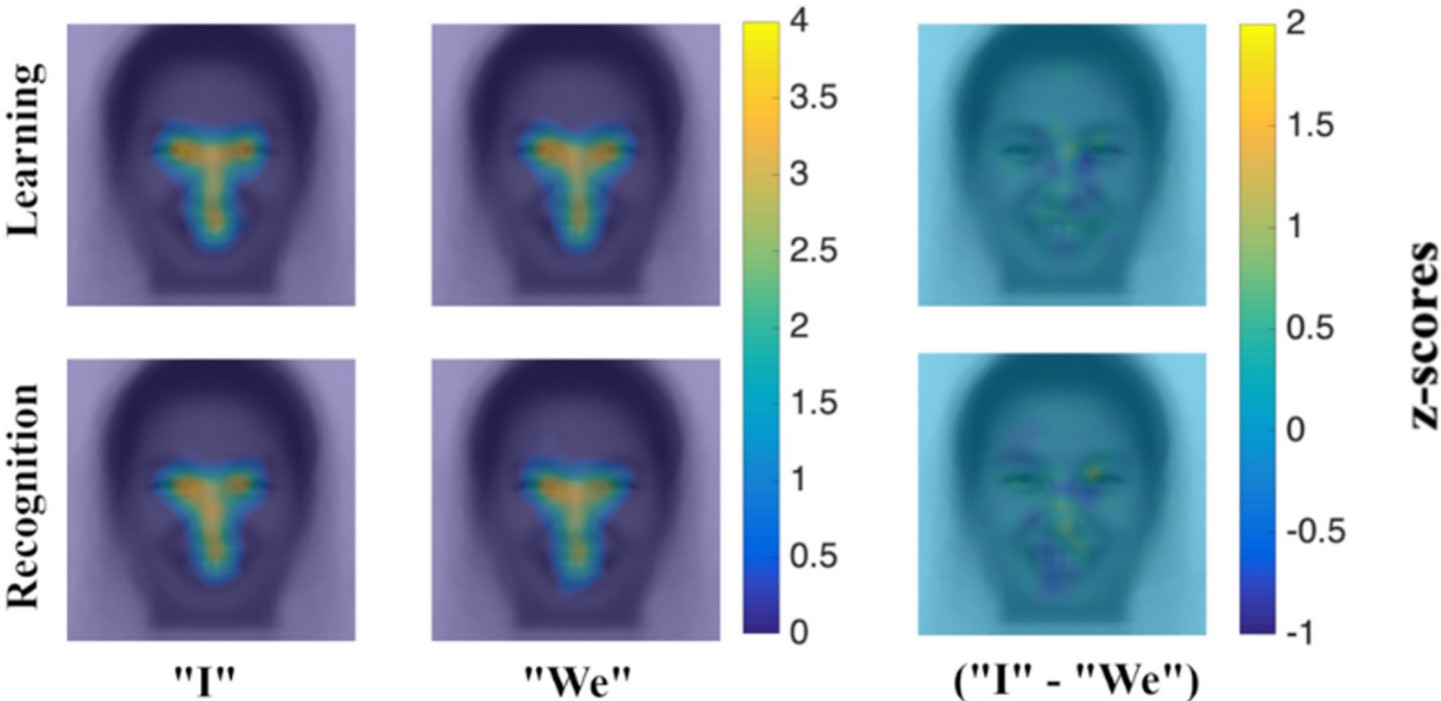

**Fig 4. Fixation maps–priming differences.** First column illustrates the "I" personal pronoun priming group's average fixation maps for learning and recognition periods respectively; second column illustrates the "We" personal pronoun priming group's average fixation maps for learning and recognition periods respectively; third column illustrates the "I" priming group (yellow) and "We" priming group (blue) fixation biases for learning and recognition periods respectively (no significant priming group differences revealed by iMap4).

AICS, and ninety-two of them (forty-six Canadians and forty-six Chinese) completed the HVICS. Our aim for this experiment was a minimal statistical power of 80% given a medium effect size (r = .3) and α = .05. This effect size was based on previous studies that found large effect sizes for cultural differences in face recognition [13, 46]. Here, we also used a smaller effect size than what was found in those studies. According to a G*Power assessment for a Pearson's correlation (bivariate normal model), the total sample size required would be 84.

## Material and stimuli

Images were 490×500 pixels in size and displayed with a white screen-wide background on a 1920×1080 pixel (53×30 cm) BenQ GL2250 LCD monitor (refresh rate of 100 Hz). On screen, faces had an average width of 8.5 cm. Participants were seated at a viewing distance of 57 cm from the computer screen, face stimuli subtending approximately 9 degrees of visual angle horizontally. The face dataset was the same as in Experiment 1, but this time participants only viewed own-race faces. This decision was made based on the absence of a significant effect of face ethnicity in the original Blais et al. [13] article.

## Eye-tracking apparatus

Eye movements were recorded at a sampling rate of 1000 Hz with the SR Research Desktop-Mount EyeLink 1000 eye-tracker; only the dominant eye was tracked (determined using the same procedure as in Experiment 1). To ensure that viewing distance was constant throughout the entire experiment and that eyes were accurately tracked, participants were asked to

**Table 1. Mixed ANOVAs with culture x priming on ROI (fixation percentages).**

Learning period

| | | Main effect of Culture | | | |
|---|---|---|---|---|---|
| ROI | Mean difference (%) | 95% CI | F (1, 34) | p | η2p |
| Eyes | 7.34 | [-3.12, 17.80] | 2.030 | .163 | .056 |
| Center | 1.67 | [-7.97, 11.29] | .123 | .728 | .004 |
| Mouth | .01 | [-6.91, 6.93] | .00001 | .997 | .000 |
| Contour | -9.02 | [-15.66, -2.36] | 7.590 | .009 | .182 |

| | | Main effect of Priming | | | |
|---|---|---|---|---|---|
| ROI | Mean difference (%) | 95% CI | F (1, 34) | p | η2p |
| Eyes | -.19 | [-3.60, 3.22] | .005 | .944 | .000 |
| Center | .62 | [-2.62, 3.85] | .164 | .688 | .005 |
| Mouth | 1.84 | [-0.91, 4.59] | 1.260 | .270 | .036 |
| Contour | -2.27 | [-5.53, 099] | 1.934 | .173 | .054 |

| | | Culture x Priming interaction | | | |
|---|---|---|---|---|---|
| ROI | -- | -- | F (1, 34) | p | η2p |
| Eyes | -- | -- | 1.193 | .282 | .034 |
| Center | -- | -- | .030 | .863 | .001 |
| Mouth | -- | -- | 2.080 | .158 | .058 |
| Contour | -- | -- | .013 | .911 | .000 |

Recognition period

| | | Main effect of Culture | | | |
|---|---|---|---|---|---|
| ROI | Mean difference (%) | 95% CI | F (1, 34) | p | η2p |
| Eyes | 4.03 | [-7.47, 15.53] | .507 | .481 | .015 |
| Center | 8.38 | [-3.81, 20.56] | 1.950 | .171 | .054 |
| Mouth | -2.89 | [-12.34, 6.56] | .386 | .538 | .011 |
| Contour | -9.52 | [-15.88, -3.15] | 9.230 | .005* | .213 |

| | | Main effect of Priming | | | |
|---|---|---|---|---|---|
| ROI | Mean Difference (%) | 95% CI | F (1, 34) | p | η2p |
| Eyes | 2.07 | [-1.53, 5.67] | 1.220 | .277 | .035 |
| Center | 1.77 | [-1.76, 5.30] | .745 | .394 | .021 |
| Mouth | -1.33 | [-4.93, 2.26] | .245 | .623 | .007 |
| Contour | -2.51 | [-6.48, 1.46] | 1.773 | .192 | .050 |

| | | Culture x Priming interaction | | | |
|---|---|---|---|---|---|
| ROI | -- | -- | F (1, 34) | p | η2p |
| Eyes | -- | -- | .036 | .850 | .001 |
| Center | -- | -- | .690 | .412 | .020 |
| Mouth | -- | -- | 2.509 | .122 | .069 |
| Contour | -- | -- | .232 | .633 | .007 |

*p < .006

The first column is the ROI; the second column is the mean difference of fixation percentage on each ROI between groups (Swiss–Chinese for the main effect of culture; I–We for the main effect of priming condition); the third column is the 95% confidence interval around the mean difference; the last three columns are results of the ANOVA.

position their head on a chin and forehead rest, facing the screen at the appropriate viewing distance.

A nine-point manual calibration of ocular fixations was performed before each task block (see protocol section for details regarding "blocks"). Calibrations were then validated with the EyeLink software and repeated when necessary until the optimal calibration criterion was reached. At the beginning of each trial, participants were instructed to fixate a dot at the center of the screen to perform a drift correction. The experiment was implemented in MATLAB R2012b, using the Psychophysics [51–53] and EyeLink [54] toolboxes (see http://psychtoolbox.org).

## Protocol

The same Old/New task as in Experiment 1 was used (see Fig 1), except that there was no priming text to read. At the end of this experiment, participants completed the AICS and HVICS.

## Analyses

The AICS and the HVICS were analyzed separately. We standardized questionnaire data to minimize cultural bias in responses, as recommended by Fischer [59]. Z-scores were produced for each questionnaire item by using participants' individual overall mean and standard deviation across one questionnaire. Then, for each participant separately, standardized questionnaire items were grouped into their respective individualism or collectivism subdimensions for each questionnaire.

For the AICS, scores were averaged along three separate individualism subdimensions: (1) "Competition", which measures how much one values and enjoys competition (2) "Uniqueness", which measures how much one values their individuality and unique preferences, and (3) "Responsibility", which measures how much one values independence, self-reliance, individual responsibility and making their own decisions; as well as two separate collectivism subdimensions: (4) "Advice", which measures how much one is inclined to consult with relatives or close friends before making a personal decision, and (5) "Harmony", which measures how much one values courtesy and in-group cohesion over personal opinion.

For the HVICS, scores were averaged along two separate individualism subdimensions: (1) "Horizontal Individualism" (HI), which measures how much one values personal identity, independence and self-reliance, and (2) "Vertical Individualism" (VI), which measures how much one values competition and enjoys winning over other; as well as two separate collectivism subdimensions: (3) "Horizontal Collectivism" (HC), which measures how much one's pride and pleasure comes from other's success and well-being, and (4) "Vertical Collectivism" (VC), which measures how much one values group hierarchy and duty towards family. Then, for each questionnaire separately, we averaged together subdimensions of Individualism to compute a general Individualism score and subdimensions of Collectivism to compute a general Collectivism score.

Individual fixation maps and iMap4 LMM parameters were computed by following the same procedure as in Experiment 1. One participant's eye movement data was removed for the recognition periods because their fixation map was empty. We started by running a fixed effect ANOVA on fixation maps between cultural groups. Next, to measure the effect of social values on eye movements, we performed a linear regression on the fixation maps, with questionnaire scores as independent variables, using the "model beta" option included in the iMap4 toolbox.

As a second step, we investigated the effect of culture on specific face regions where significant differences were previously found between West Europeans/North Americans and East Asians. We conducted a ROI analysis following the same procedure as in Experiment 1. Percentage of fixations allocated to the eyes, nose, mouth, and rest of the face respectively were compared between both cultural groups (Canadians and Chinese). To examine group differences on fixation percentages for each ROI, we ran bilateral Independent Samples t-tests between cultures on each ROI separately. The bilateral significance threshold (p < .025) was corrected for multiple comparisons using a Bonferroni correction, where differences on each ROI are tested at p = .025/4 = .006.

Next, across all participants, regardless of culture, correlations between percentage of fixations and each questionnaire subdimension were performed for each ROI separately. The skipped-correlation algorithm from the Robust Correlation Toolbox [60] was initially applied for a robust correlation with multiple comparison correction across questionnaire subdimensions. Pairs of subdimensions and ROI that showed a significant correlation were then subjected to a partial correlation with "participant's culture" as co-variable.

## Results

### Questionnaires

Questionnaire z-scores were plotted for AICS (Fig 5A) and HVICS (Fig 5B) along the general dimensions of Individualism and Collectivism, as well as each subdimension of the latter constructs, distinctly for Canadian and Chinese participants.

For the AICS, Independent Samples t-tests between Canadian and Chinese participants show that there are no significant group differences for individualism [t(113) = -1.020, p = .310, Cohen's d = -.191] and collectivism [t(113) = 1.020, p = .310, Cohen's d = .191]. Independent Samples t-tests between Canadian and Chinese participants for each subdimension show that Canadians score significantly higher on "Unique" [t(113) = 5.094, p < .001, Cohen's d = .954], "Responsibility" [t(113) = 3.730, p < .001, Cohen's d = .699], and "Harmony" [t(113) = 5.314, p < .001, Cohen's d = .996], whereas Chinese score higher on "Competition" [t(113) = -8.689, p < .001, Cohen's d = -1.630] and "Advice" [t(113) = -3.687, p < .001, Cohen's d = -.691]. It is interesting to note that Canadians do not score higher on all conventionally individualistic dimensions (i.e., "Competition", "Unique", and "Responsibility"), as one would expect, and that Chinese do not score higher on all conventionally collectivistic dimensions (i.e., "Advice" and "Harmony"), as one would also expect. Notably, these results corroborate the data presented by Shulruf et al. [40] to validate the usefulness of the selected subdimensions to better discriminate cultural groups.

For the HVICS, Independent Samples t-tests between Canadian and Chinese participants show that Chinese participants' ratings for Individualism are significantly higher than those of Canadian participants [t(97) = -5.239, p < .001, Cohen's d = -1.060], whereas Canadian participants' ratings for Collectivism are significantly higher than those of Chinese participants [t(97) = 5.214, p < .001, Cohen's d = 1.050]. Independent Samples t-tests for each subdimension reveal no significant difference between cultural groups for HI [t(97) = -.264, p = .793, Cohen's d = -.053] and for VC [t(97) = -1.511, p = .134, Cohen's d = -.305]. However, Chinese participants score significantly higher on VI [t(97) = -6.837, p < .001, Cohen's d = -1.378]; Canadian participants score significantly higher on HC [t(97) = 9.474, p < .001, Cohen's d = 1.909]. These results show that Chinese participants' overall higher ratings for Individualism in the HVICS are mostly driven by VI; and Canadian participants' higher ratings for Collectivism are mostly driven by HC.

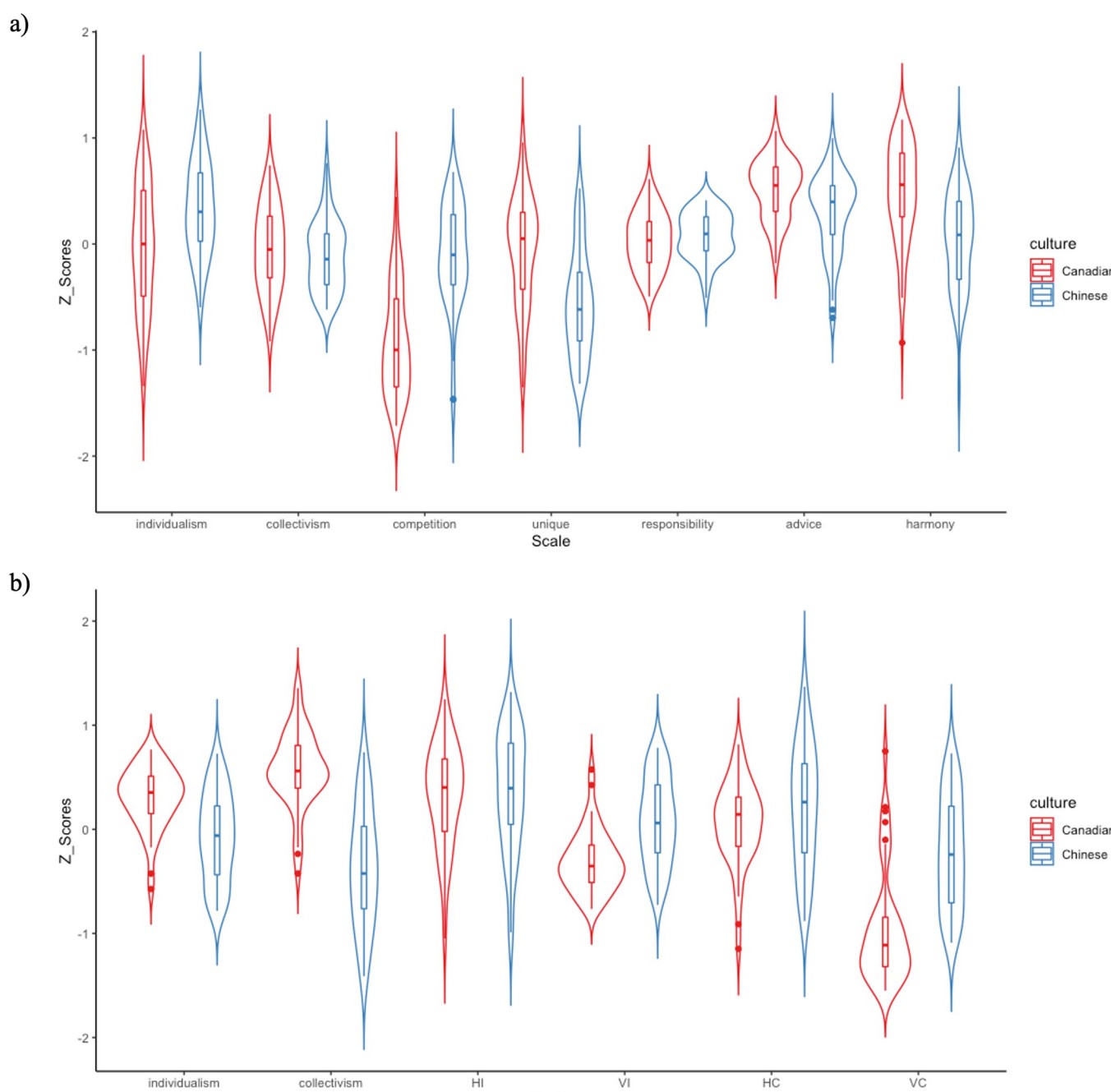

**Fig 5. Questionnaire z-score distributions.** a) Z-score distributions for Canadian and Chinese participants on IND/COL general dimensions and subdimensions of AICS; b) Z-score distributions for Canadian and Chinese participants on IND/COL general dimensions and subdimensions of HVICS.

The cultural differences on dimensions VI and HC confirm the results obtained on dimensions "Competition" and "Harmony" respectively. Indeed, VI is conceptually similar to "Competition" and HC is conceptually similar to "Harmony". VC, which is conceptually similar to "Advice", is higher for Chinese participants. As opposed to "Advice", however, VC more clearly emphasizes group hierarchy. No significant difference was found for the dimension HI,

which could be understood as combining "Unique" and "Responsibility", and thus remains more broadly defined.

### Culture and eye movements

Fixation maps (see Fig 6) reveal the typically observed cultural differences in fixation patterns (more fixations on the eyes for West Europeans; more fixations on the center for East Asians), especially for learning periods. Significantly, iMap4 reveals a left eye fixation bias for Canadian participants during learning and recognition periods (p < .05). Above and around the right eye there appears to be a Chinese fixation bias, which is unexpected and might be linked to our Chinese participants' tendency to move their gaze around the face in a diffuse manner. This pattern also appears above the eyes on fixation maps in Experiment 1, albeit not as significantly (see Fig 3). Independent Samples t-tests on ROI confirm significantly more fixations toward the eyes for Canadian observers during learning [t(110) = 4.386; p < .001; Cohen's d = .835] and recognition [t(109) = 2.723; p = .008; Cohen's d = .5206]. This effect is stronger for learning periods, compared to recognition periods. By contrast, they confirm significantly more fixations around the face (rest of the face ROI) for Chinese observers during learning [t(110) = -5.509; p < .001; Cohen's d = 1.049] and recognition [t(109) = -3.916; p < .001; Cohen's d = .749]. However, as can be seen on Chinese participants' group maps (Fig 6; second column), relatively less fixation time is spent around the face than in the center as indicated by the bluish hue (lower z-scores) at the periphery and orange color (higher z-scores) near the center. Average fixation percentage differences between cultures are reported in Table 2 for each ROI.

In order to evaluate whether this difference was due to poorer data quality, we measured the mean percentage of data loss per trial for every participant in both the Canadian and

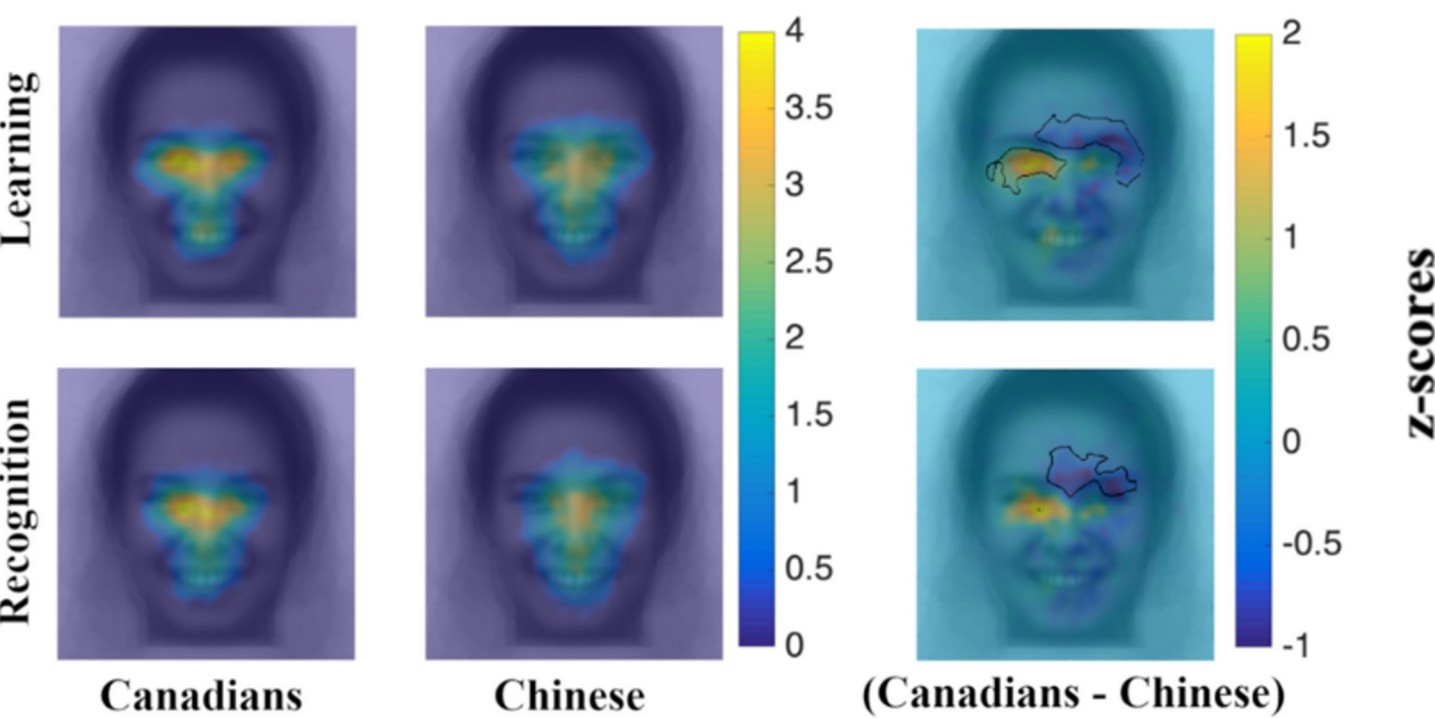

**Fig 6. Fixation maps.** First column illustrates Canadian participants' average fixation maps for learning and recognition periods respectively; second column illustrates Chinese participants' average fixation maps for learning and recognition periods respectively; third column illustrates Canadian (yellow) and Chinese (blue) fixation biases for learning and recognition periods respectively (significant cultural differences revealed by iMap4 are delineated in black).

**Table 2. Independent t-tests with culture on ROI (fixation percentages).**

Learning period

| | Main effect of Culture | | | | |
|---|---|---|---|---|---|
| ROI | Mean difference (%) | 95% CI | t (110) | p | Cohen's d |
| Eyes | 8.91 | [4.89, 12.94] | 4.386 | < .001* | 0.835 |
| Center | 1.17 | [-3.14, 5.47] | 0.537 | .592 | 0.102 |
| Mouth | 1.54 | [-1.24, 4.32] | 1.099 | .274 | 0.209 |
| Contour | -11.62 | [-15.80, -7.44] | -5.509 | < .001* | 1.049 |

Recognition period

| | Main effect of Culture | | | | |
|---|---|---|---|---|---|
| ROI | Mean difference (%) | 95% CI | t (110) | P | Cohen's d |
| Eyes | 6.40 | [1.74, 11.05] | 2.723 | .008 | 0.521 |
| Center | 4.23 | [-0.65, 9.11] | 1.718 | .089 | 0.329 |
| Mouth | -0.87 | [-4.77, 3.03] | -0.443 | .659 | 0.085 |
| Contour | -9.76 | [-14.70, -4.82] | -3.916 | < .001* | 0.749 |

*p < .006

The first column is the ROI; the second column is the mean difference of fixation percentage on each ROI between groups (Canadians–Chinese); the third column is the 95% confidence interval around the mean difference; the last three columns are results of the Independent Samples t-tests.

Chinese samples. A paired t-test revealed that there was a significantly higher percentage of data loss in the Chinese sample (t(97) = 5.235, p < .001, Cohen's d = 1.052). However, there is no significant difference in error after calibration between the two samples (t(97) = 1.842, p = .069, Cohen's d = .370), with error being slightly higher on average in the Canadian sample (.542 degrees) compared to the Chinese sample (.442 degrees). As such, there is little chance our results are caused by poor data quality, even though higher data loss in the Chinese sample is regrettable. These analyses could not be run for experiment 1, due to the format data was saved in not including measures of data quality.

## Social values and eye movements

iMap4's *Model Beta* LMM analysis did not yield any significant results for social values, suggesting that any effect of social values on fixation patterns might not be very robust, compared to the effects of culture revealed by iMap4. For this LMM analysis we chose a significance threshold of .05, and applied a bootstrap procedure for multiple comparison correction. Notwithstanding, we considered it worthwhile to examine correlations between social values and ROI since iMap4 remains a very conservative tool. Analyses with important face regions, as opposed to data-driven pixel-by-pixel tests, will provide useful insights. Since our cultural groups do not seem to be respectively defined by either individualism or collectivism as conceptualized by our questionnaires, significant correlations between those social constructs and ROI, as predicted by the social orientation hypothesis, would mean that those constructs are better associated with fixation distribution than the cultural groups themselves. Obviously, since group differences were observed, we expect simple correlations to be significant. However, partial correlation analyses can provide more information. The following paragraphs will only discuss significant correlations and partial correlation analyses, but all correlations are reported in the Supplementary material (S1 to S4 Tables in S1 File).

For the AICS, we found significant correlations between individualism and central fixations (r = .279; p < .01), and collectivism and central fixations (r = -.280; p < .01). These results were consistent for learning and recognition periods (partial correlations for recognition

periods: for IND, r = .071, p = .459; for COL, r = -.071, p = .459). However, these become non-significant when including culture in a confirmatory partial correlation (partial correlation for IND: r = .119; p = .212, and partial correlation for COL: r = -.119; p = .212).

For the HVICS, there was a significant correlation between fixations to the rest of the face and individualism as well as collectivism. Individualism was associated with more fixations around the face (r = .290; p < .01), whereas collectivism was associated with less fixations around the face (r = -.290; p < .01). These results were only significant for learning periods, but a similar trend appears for recognition periods. Interestingly, these correlations are symmetrically opposed despite the fact that individualism and collectivism are measured as two separate constructs. Confirmatory partial correlations with participants' culture as co-variable, however, shows the effect disappears when accounting for culture (partial correlation for IND: r = .066; p = .534, and partial correlation for COL: r = -.066; p = .534).

Then, we proceeded to examine the link between ROI and subdimensions of IND/COL, for both questionnaires separately. For the AICS, after multiple comparison correction, only "Competition" was significantly negatively correlated with the eyes during learning periods (r = -.460; p < .001). By contrast, "Harmony" was significantly positively correlated with the eyes (r = .384; p < .001). Only "Competition" remained correlated with eyes regardless of culture (partial correlations: for competition r = -.169, p = .076; for harmony r = .127, p = .183). For recognition periods, only a positive correlation between "Competition" and the rest of the face was significant (r = .286; p < .01). However, partial correlation shows that this was rather explained by cultural differences on fixation bias (r = -.135; p = .159).

For the HVICS, "VI" (similar to Competition) was again significantly negatively correlated with the eyes during learning periods (r = -.379; p < .001), and "HC" (similar to Harmony) was still significantly positively correlated with the eyes (r = .315; p < .01). "VI" remained correlated even when controlling for culture (partial correlation: r = -.238; p = .023), but correlation with "HC" was better explained by culture (partial correlation: r = .073; p = .491), that is by Canadians scoring higher on "HC" and fixating more the eyes than Chinese participants. Significant negative correlation between VI and eyes, regardless of culture, confirms results obtained with "competition" during learning periods. By contrast, VI was positively correlated with the rest of the face (r = .289; p < .01), whereas HC was negatively correlated with that area (r = -.391; p < .001). However, partial correlations suggest that this was better explained by the fact that Chinese participants score higher on VI, lower on HC, and fixate more that area of the face (partial correlations: for VI, r = -.002, p = .989; for HC, r = -.033, p = .753). During recognition periods, only VI remained negatively correlated to the eyes (r = -.320; p < .01) and positively correlated to the rest of the face (r = .296; p < .01). Partial correlations indicate again that VI remains negatively correlated to eyes regardless of culture (r = -.249; p = .018), but that culture better explains the positive correlation between VI and the rest of the face (r = .057; p = .595).

## Discussion

First, according to our data, it is not possible to assert that our sample of Canadian participants is purely "individualistic", nor that our sample of Chinese participants is purely "collectivistic". Rather, our participants adhere more or less to subdimensions of individualism and collectivism in a way that is culturally distinguishable. Second, our results indicate that cultural differences related to fixation distributions cannot be unequivocally explained by individual variations along individualism and collectivism. Partial correlations with culture as covariable show that most significant associations between ROI and social values seem to be explained away by cultural membership. It is plausible that some individual social tendencies might be

associated with particular fixation biases. There was a persistent negative association between fixation duration toward the eyes and scores on the "competition" subdimension (AICS), as well as the similar "VI" subdimension (HVICS), that remained regardless of culture.

## General discussion

Face perception plays a crucial role in social development. Discovering how and why it might develop differently from one culture to another is an important challenge for our globalized world and modern multicultural social interactions. The present study aimed to investigate the suitability of the social orientation hypothesis as potential explanation for the substantial cultural differences characterizing face perception. In fact, the social orientation hypothesis has been posited as a prominent explanation for perceptual and cognitive cultural differences in many studies [22, 61]. In light of empirical evidence showing cultural differences in eye movements during face recognition between East Asians and West Europeans/North Americans, we probed the idea that social orientation could also explain cultural differences in face perception. In our study, we used an Old/New face recognition task and two different experimental protocols to measure social orientation with a full cross-cultural design.

## Cultural differences

In both of our experiments, our data revealed cultural contrasts in visual attention between West Europeans/North Americans and East Asians. We revealed a higher proportion of fixations toward facial features, especially towards the eyes, for West European and North American participants, and fewer fixations toward those features for East Asian participants. However, instead of displaying the usual central fixation bias observed with East Asian populations, our Chinese participants instead fixated more significantly around the face, but not on facial features. This finding is inconsistent with previous studies. Analyses on data quality suggest this is not due to poorer data quality in the Chinese sample, at least for experiment 2. This behavior could qualitatively and conceptually remain in line with common East Asian holistic/global attentional patterns, and is consistent with strategies employed during scene perception [30]. East Asian observers adopt a visual strategy favoring the extra-foveal processing of facial features. We replicated this pattern in Experiments 1 and 2 of our studies, despite testing two independent groups of observers for each culture. It is worth noting that Chinese participants recruited for the present study did not necessarily come from the same region as East Asian participants from previous studies [13, 14, 38]. Increased arousal due to exposure to urban environments has been proposed as a factor that could drive cultural differences [62], since exposure to urban environments induces a more global bias in perception [63]. Therefore, it is possible that East Asians coming from different areas develop diverse eye movement strategies to deploy broad attention and extra-foveal processing of facial features [64].

It is worth noting that recent studies have also put forward within-group variations during face perception. In fact, individual differences within the same cultural group have specifically been found with respect to eye movements during face recognition tasks. For instance, Miellet et al. [65] show that West European observers can alternate between a more "local" or "global" strategy to recognize a celebrity identity, as a function of the location on which their first fixation lands. Chuk et al. [66, 67] revealed individual eye movement patterns that can be classified as being either "holistic" or "analytic", emphasizing the existence of individual differences within the same group of East Asian observers. Individual differences in face processing strategies are shown to be remarkably stable and reflect individual optimal gaze positions for recognition [9, 68, 69].

Cultural variations in eye movements should be seen as variations on a continuum spanning from local to global strategies, with the probability of West Europeans/North Americans and East Asians appearing more at one extremity or the other of the continuum. West European, North American and East Asian cultures represent very complex systems of a manifold of interacting elements, all shaping cognition in one way or another. Nevertheless, distinctive group patterns of facial recognition continue to emerge. These general differences that persist between certain populations are interesting phenomena that require further exploration and might even give us a hint on what drives individually motivated patterns of fixations. A series of potential mechanisms underlying cultural differences in perception have been recently put forward in the literature [70].

## Social orientation

In Experiment 1, we failed to reveal a significant effect of social orientation priming on eye movements, which is consistent with the recent results reported by Liu et al. In Experiment 2, we also failed to reveal a clear association between social orientation questionnaires and eye movements. This is consistent with previous results by Lacko et al. [71], who showed that social orientation did not predict changes in processing of different stimuli such as maps or Navon letters. The fact that no significant results were obtained from two independent samples collected in studies with vastly different experimental paradigms makes it unlikely for these observations to be the result of a methodological flaw. While we observed some associations between eye movements and IND/COL constructs, these results were mostly driven by participants' culture rather than social orientation per se.

Interestingly, in Experiment 2, our Chinese participants rated higher for individualism and lower for collectivism than our Canadian participants. This trend is inconsistent with results from previous studies that considered global individualism/collectivism and was mostly driven by the subdimension "competition" in the AICS questionnaire and the corresponding subdimension "VI" in the HVICS. As was mentioned earlier, the higher scores on the individualism subdimension "competition" in East Asians is consistent with the results obtained by Shulruf et al. [40]. When IND/COL subdimensions were taken into account, we only found significant negative correlations between the eyes and subdimensions "competition" and "VI", regardless of participants' culture. These correlations might reflect more complex and nuanced effects of social values on visual perception.

A possible explanation for why our data contradicts the literature on this matter might stem from how individualism is defined in the current questionnaires. There is a debate about whether or not individualism and competitiveness should be put under the same umbrella [72]. Evidence from cross-cultural management research points to a decoupling of those concepts. For instance, during an experimental cooperation game, Chinese participants that are more individualistic cooperate further than those that are more collectivistic [73]. It is suggested that this behavior might be dependent on the reference group, namely whether cooperative action is directed towards ingroup or outgroup members. It is plausible to assume that collectivistic societies can be competitive for reasons that are collective, rather than personal. Accordingly, in the current study, we can observe that our Chinese participants do indeed score higher on competition (and VI) than our Canadian participants, however it might be erroneous to strictly assume that they are more individualistic than our Canadian participants, as they do not score higher than Canadians on other individualism subdimensions.

Alternatively, we also cannot exclude the possibility that our sample of Chinese participants was more individualistic than average, suggesting that it might be worthwhile to revise our assumptions of China as purely "collectivistic" and Canada as purely "individualistic." Indeed,

the way we formally define what constitutes an individualistic or collectivistic society does become less clear-cut over time. The deconstruction notions of individualism and collectivism emphasizes the importance of using questionnaires with subdimensions to investigate these concepts. In the context of our study, general notions of individualism and collectivism aside, our results indicate the existence of an association between eye movements for face recognition and individual levels of competitiveness. This link is interesting and worth further investigation, provided that we conceptualize "competition" in a way that reflects both its individualistic and collectivistic dimensions.

However, due to the many unexpected results that were found in our study, we should be careful in our interpretations. The fact that the eye-movement patterns we found were inconsistent with previous studies, and that the priming paradigm may not have been successful in causing differences to emerge make the interpretation of our results difficult. More research is needed to fully understand cultural expressions of individualism and collectivism across different countries before studying this topic further. That being said, we did not manage to reveal an effect of social orientation on eye movements with methods that were consistent with what was previously used in social orientation research. Further replications using more controlled experimental parameters, measures of success in the priming task and more behavioral measures of social orientation would benefit immensely and give valuable insight to the field.

## Limitations

The results presented here provide undeniable evidence that more studies should be conducted before assuming the existence of a direct link between social values and eye movement patterns. Nonetheless, the current study does present some noteworthy limitations, namely that it relies on the interpretation of null results. That being said, we aim to provide well documented hypotheses and careful conclusions to steer future research in the right direction.

First, some limitations concerning statistical power in our experiments need to be considered. While we calculated minimum sample size by using effect sizes reported in previous studies, these studies all had a relatively low number of participants. This is not recommended, as effect sizes detected on smaller samples tend to be overestimated [74]. Also, in experiment 1, we computed a minimum sample size of 34 participants. However, this assumes equal sample sizes between groups, yet we only have 15 Swiss participants. This is below the recommended sample size and may prevent us from detecting the effect. Taken together, it is a possibility that experiment 1 may have been underpowered, making it more difficult to reveal an effect. That being said, we believe that experiment 2 alleviates this limitation as it has a bigger sample size and as such should not be underpowered.

With respect to measures of social orientation, while the questionnaires used in Experiment 2 were extensively validated in prior studies, questionnaire research for cultural psychology as a whole has some serious limitations. While it has been widely used to study varied phenomenon, many potential biases can affect validity (ex. construct inequivalence, item equivalence, response biases, etc.) As shown by Lacko et al. [75], analyzing while maximizing measurement invariance on a questionnaire can lead to radically different interpretations than simply analyzing the questionnaire in a "traditional" fashion. From their findings, concerns arise that factor structure may not hold when comparing full questionnaires between cultures, as even though studies have shown factor structure to be preserved [76], reported model parameters are much lower than average. In this study, we wished to use comparable measures as what was used before, so we opted for questionnaires. With these measures, we do not manage to replicate previous findings. Studies focusing on the measure of social orientation itself should

be done to better understand how to efficiently measure this concept, and exploring how consistent results are between studies could ensure better comparability.

We recommend exploring different measures for future studies, perhaps more behavioral in nature, to ensure better measurement validity and comparability between cultures. Research by Talhlem and collaborators [77–79] observe participants' behavior in controlled contexts to make inferences, instead of asking questions via questionnaire. While this can complexify data collection, it remains a promising avenue for cultural research.

The pronoun-circling method we used in Experiment 1 is meant to prime "self-construal [55]". However, a common problem remains whether we evaluate interdependence or collectivistic values: we must define what the "we" in collectivism stands for. For example, across several scales to measure collectivism, depending on the content of the questions, Americans are sometimes more or less collectivistic than Japanese or Koreans [80, 81]. Accordingly, several studies use a tripartite model of self-construal including the individualistic personal pronoun ("I"), the relational "we", and the collective "we" [55]. A recent priming study using this tripartite model shows that Asians' social identity is primed more with an interpersonal/relational scenario, whereas Australians' social identity is primed more with a scenario involving larger groups or collectives [82]. Critically, the way ingroups are defined within a given culture might even modulate the way in which interdependence (or collectivism) relates to face recognition ability [83]. For instance, North Americans tend to construct their social identities based on abstract categories such as nationality. Consequently, when they are more interdependent, they are better at recognizing novel own-race faces, but not other-race faces. East Asians tend to identify with smaller tight-knit groups that have more defined boundaries and emphasize personal relationships. Consequently, East Asians that are more interdependent are worse at recognizing any novel face.

Another limitation is that we could not run data quality analyses for experiment 1. Sadly, due to the data was saved in, we retained no information relevant for data quality. That being said, calibration quality was monitored all throughout experiment 1, and as such we are confident the data is of good quality.

Finally, there is a possibility that the priming task used in Experiment 1 was not effective. We had no way of knowing whether the priming was effective on participants, and it is possible that a within-subject design was not appropriate to detect differences in eye-movement patterns. However, it is crucial to note that the pronoun circling manipulation used here has previously lead to many positive results with a variety of tasks [34, 36, 84, 85]. In addition, the current study replicates the absence of a social orientation priming effect on eye movements reported in Liu et al. [36], using the same priming manipulation and face recognition task, but using a between-participant design. While a metanalysis by Oysermann & Lee [86] has shown that priming of collectivism and individualism successfully caused changes in cognition across many studies, priming paradigms have been criticized by many authors recently due to poor replicability. Different definitions of what constitutes priming "social" constructs [87] as well as differences in study design [39] can lead to different conclusions when evaluating the replicability of these studies. We believe that in-depth research on the mechanisms behind individualism/collectivism priming effects, as well as the replicability of priming studies, is necessary before further examining the question using priming paradigms.

## Conclusion and future directions

In conclusion, our findings indicate that social orientation, at least when defined by pure individualism/collectivism, does not unequivocally explain differences in face recognition strategies. If a link exists between social orientation and visual strategies for facial recognition, it is

at the very least not a straightforward one. Indeed, our results suggest other factors are at play, possibly interacting with social orientation, to form the truly complex and dynamic web that is culture. As a matter of fact, one's environment can be viewed as an assemblage of cultural, social, biological, and geographical constraints.

Other factors of our visual environment should also be considered as complementary or alternative explanations for the observed differences in facial recognition strategies. In order to truly understand the way culture interacts with perceptual development and facial recognition, future studies should explore diverse aspects of one's environment including geographical elements such as the visual configuration of the surroundings (e.g., urbanization [63]), as well as individual elements such as biological conditions (e.g., myopia, a visual condition more prevalent in East Asian countries [88, 89]). All in all, the genuine visual and social mechanisms at the root of cultural differences in face processing still remain to be defined.

## Supporting information

**S1 File.**
(DOCX)

## Acknowledgments

We thank Meike Ramon for her comments and work on a previous version of the manuscript.

## Author Contributions

**Conceptualization:** Daniel Fiset, Caroline Blais.

**Data curation:** Amanda Estéphan, Caroline Blais.

**Formal analysis:** Amanda Estéphan, Caroline Blais.

**Funding acquisition:** Daniel Fiset, Caroline Blais.

**Investigation:** Amanda Estéphan, Daniel Fiset, Caroline Blais.

**Methodology:** Amanda Estéphan, Daniel Fiset, Caroline Blais.

**Project administration:** Daniel Fiset, Caroline Blais.

**Resources:** Daniel Fiset, He Lingnan, Roberto Caldara, Caroline Blais.

**Software:** Amanda Estéphan, Caroline Blais.

**Supervision:** Daniel Fiset, Caroline Blais.

**Validation:** Caroline Blais.

**Visualization:** Amanda Estéphan, Caroline Blais.

**Writing – original draft:** Francis Gingras, Amanda Estéphan, Daniel Fiset, Caroline Blais.

**Writing – review & editing:** Francis Gingras, Daniel Fiset, He Lingnan, Roberto Caldara, Caroline Blais.

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
