## [Decision Letter · Decision Letter 0]

1 Jun 2023

PONE-D-23-11102Cultural Differences in Eye Movements for Face Recognition are not Modulated by Social OrientationPLOS ONE

Dear Dr. Blais,

Thank you for submitting your manuscript to PLOS ONE. After careful consideration, we feel that it has merit but does not fully meet PLOS ONE’s publication criteria as it currently stands. Therefore, we invite you to submit a revised version of the manuscript that addresses the points raised during the review process. In particular, as you noted in your cover letter, the word "culture" should be used with care. You need to clearly state which cultural features are probed with your design.

Also, Figure 5 is hard to interpret, you use bar graphs which hide the underlying distribution of the data, see https://garstats.wordpress.com/2016/03/09/one-simple-step-to-improve-statistical-inferences/

Please consider using an alternative such as violin plots, raincloud plots, or even simply add individual data points.

We look forward to receiving your revised manuscript.

Kind regards,

Antoine Coutrot

Academic Editor

PLOS ONE

Reviewers' comments:

Reviewer's Responses to Questions

**Comments to the Author**

1. Is the manuscript technically sound, and do the data support the conclusions?

Reviewer #1: Yes

Reviewer #2: No

Reviewer #3: Partly

2. Has the statistical analysis been performed appropriately and rigorously? 

Reviewer #1: Yes

Reviewer #2: Yes

Reviewer #3: Yes

3. Have the authors made all data underlying the findings in their manuscript fully available?

Reviewer #1: Yes

Reviewer #2: Yes

Reviewer #3: No

4. Is the manuscript presented in an intelligible fashion and written in standard English?

Reviewer #1: Yes

Reviewer #2: Yes

Reviewer #3: Yes

5. Review Comments to the Author

Reviewer #1: The article is generally well written. I have no comments related to the Introduction that summarizes the previous research and the rationale of the current research well.

Comments:

1) I think it is unnecessary to repeat the same information for the both Experiments (mainly relevant for Material and Stimuli and Apparatus sections. This informatiojn might be mentioned before sections focused specifically on both experiments (and pointing out possible deviations of experimantal conditions in E1 and 2). It would shorten the text of the paper and improve its readability.

2) p. 7, participants: I would suggest reporting full G*Power settings including alpha. I was able to replicate the power analysis in G*Power but I was guessing with some settings.

The following three points are mainly my comments that are related to obstacles that we encountered in this line of research in the past and to the broader theory linking social orientation to cognition. It is up to the authors whether or not they want to include them in the paper, although I believe that a deeper discussion of these topics could improve the quality of the paper.

3) To be honest, I am not surprised that there was no main effect of priming found in study 1. Our own experiments on priming individualism always failed to cause any measurable differences in cognition to such an extent that I dont believe that simple semantic priming of this type can elicit any changes in cognition despite the results of metaanalysis conducted by Oyserman and Lee (2008). Recently, priming was critisized from left and right. it might be useful to reflect this debate (Sherman, 2021).

4) The lack of relationship between social orientation and facial cognition replicates the results of other cognitive style researches using different stimuli such as maps, or Navon Figures (Lacko, 2020).

5) Moreover, I believe that the classic questionnaires that measure INDCOL/self-construals dont hold their factor structure outside Western countries unless they are reduced to minimum number of items (Lacko, 2022) or extended to n-dimensions. I believe that despite the study conducted by Vignoles et al. (since it reports final model parameters far below generally accepted thresholds in MGCFA invariance research).

References:

Oyserman, D., & Lee, S. W. (2008). Does culture influence what and how we think? Effects of priming individualism and collectivism. Psychological bulletin, 134(2), 311.

Sherman, J. W., & Rivers, A. M. (2021). There’s nothing social about social priming: Derailing the “train wreck”. Psychological Inquiry, 32(1), 1-11.

Lacko, D., Šašinka, Č., Čeněk, J., Stachoň, Z., & Lu, W. L. (2020). Cross-cultural differences in cognitive style, individualism/collectivism and map reading between Central European and East Asian University students. Studia Psychologica, 62(1), 23-43.

Lacko, D., Čeněk, J., Točík, J., Avsec, A., Đorđević, V., Genc, A., Haka, F., Šakotić-Kurbalija, J., Mohorić, T., Neziri, I., & Subotić, S. (2022). The Necessity of Testing Measurement Invariance in Cross-Cultural Research: Potential Bias in Cross-Cultural Comparisons With Individualism– Collectivism Self-Report Scales. Cross-Cultural Research, 56(2–3), 228–267.

Vignoles, V. L., Owe, E., Becker, M., Smith, P. B., Easterbrook, M. J., Brown, R., ... & Bond, M. H. (2016). Beyond the ‘east–west’dichotomy: Global variation in cultural models of selfhood. Journal of Experimental Psychology: General, 145(8), 966.

6) I did not go through the validation studies of AICS and HVICS. Were they validated on Swiss and Chinese populations? If not was factor structure and reliability verified? From my own experience, horizontal-vertical INDCOL structure doesnt always hold for example in postcommunist European countries.

7) Minor: You report some statistics, e.g. p-values without "0" (p < .05) while other, e.g. Cohens d with it. This could be made consistent throughout the text.

8) p. 31, suggestion to choose wider variety of questionnaires: I am not entirely sure that this will solve the problem, especially in comparisons of multiple cultural samples. The first obstacle you will encounter will be the validation of the questionnaires in countries beyond the traditional USA/Western Europe vs. China/Japan/Korea samples. Then the second obstacle will be that many of the possible cross cultural differences disappear once you aggreggate the culture means. I believe that a promising approach might be to more finely identify participant profiles within each country/culture and compare countries based on the proportion of those profiles within country level smaples (Na, 2020). Another promising way is to design more behavioral oriented measures, something in line with research conducted by Talhelm. This is my personal opinion based on the problems we encountered in our research that are very similar to the current study.

References:

Na, J., Grossmann, I., Varnum, M. E., Karasawa, M., Cho, Y., Kitayama, S., & Nisbett, R. E. (2020). Culture and personality revisited: Behavioral profiles and within‐person stability in interdependent (vs. independent) social orientation and holistic (vs. analytic) cognitive style. Journal of Personality, 88(5), 908-924.

Talhelm, T., Zhang, X., & Oishi, S. (2018). Moving chairs in Starbucks: Observational studies find rice-wheat cultural differences in daily life in China. Science advances, 4(4), eaap8469.

Reviewer #2: This manuscript is nicely written and presents sufficient details about the methodology to understand the paradigms. The study is novel as it directly investigates the widely theorised role of the social orientation (individualism/collectivism) in reported cultural differences in fixation strategies during face viewing. Chinese and Western participants were assessed on individualism and collectivism using questionnaires (exp2), and a priming paradigm (exp1). Eye-tracking was used to record fixations during face viewing (exp1&2). It is concluded that social orientation could not account for cultural differences in face viewing. While this study is novel, I have some reservations about the study interpretations.

Major points

- The social orientation questionnaires resulted in unexpected patterns, which is not an issue in itself of course. However, it brings up questions about the nature of these unexpected findings (e.g., is it due to the validity of measures, or the selected participants, or is the notion of individualism/collectivism simply ‘archaic’ as the authors also suggest, etc.). The unexpected findings and the closer examination of subscales in the context of culture give rise to a discussion in itself. This makes any conclusions regarding links with cultural differences in face fixations very difficult. It seems that a manuscript focusing on measuring social orientation and deconstructing findings inconsistent with the current literature is useful before jumping to examine links with cultural differences in face viewing.

- Exp1: why were participants primed in a within-subjects design? It would seem surprising to change eye movements so quickly after each priming condition.

- exp1: we have no way of knowing whether priming was (subjectively) achieved or not. In light of the lack of priming effects, we cannot conclude whether priming was not achieved, of priming was achieved but did not modulate eye movements. Priming did not elicit expected differential fixational viewing, and the hypothesised cultural differences in face scanning patterns were not observed. Yet, the authors conclude that ‘priming individualistic & collectivistic social values did not seem to abolish the cultural differences in eye movements for face recognition’. First, it is unclear whether these social values were successfully primed. Secondly, the cultural differences in question manifest in an unexpected pattern and any conclusion should therefore be more carefully worded.

- While some cultural differences in face scanning were observed, they were not always as expected/consistent with the literature (e.g., exp2). Again, while this is not an issue in itself, the manuscript needs to be clearer that a differential pattern was observed and discuss inconsistencies with previous findings.

- How was data quality of eye tracking data assessed (beyond drift correction, e.g., % data loss, precision, etc. during trials)? This is important to establish that the greater distribution in the Chinese group was not the result of poorer data quality. This is particularly relevant given that the faces were presented in the corner of the screen, i.e., at locations where eye tracking quality tends to drop off.

Minor points

- “here we included cross-cultural participants of West European and Chinese descent.” Rephrase: ‘included both West European and Chinese participants’, rather than ‘descent’ which has wider meaning.

Reviewer #3: In this paper, the authors investigate whether the collectivist/individualist value system is a valid/comprehensive explanation for eye-movement differences observed between East Asian and Western societies during face processing. Below are my comments:

1) Intro: Page 4 lines 4-7: I suggest also mentioning evidence that does not support the value systems hypothesis, e.g. Knox, P. C. & Wolohan, F. D. Cultural diversity and saccade similarities: Culture does not explain saccade latency differences between Chinese and Caucasian participants. PloS one 9, e94424 (2014).; Kardan, O., Shneidman, L., Krogh-Jespersen, S., Gaskins, S., Berman, M. G., & Woodward, A. (2017). Cultural and developmental influences on overt visual attention to videos. Scientific reports, 7(1), 1-16.

2) Power: In general it is not recommended to estimate effect size from previous literature that had small sample size, because those effect sizes would be very likely overestimated. For example see Button, K. S., Ioannidis, J. P., Mokrysz, C., Nosek, B. A., Flint, J., Robinson, E. S., & Munafò, M. R. (2013). Power failure: why small sample size undermines the reliability of neuroscience. Nature reviews neuroscience, 14(5), 365-376.

3) In experiment 1, the power analysis assumes equal group sizes, so the N = 15 (which is fewer than 34/2 = 17) for Swiss is below the recommended minimum sample size.

4) In experiment 1, isn’t the finding that Chinese participants fixated on ‘rest of face’ more than Swiss in the opposite of expected direction? This sentence in page 15 seems like moving the goalposts a bit to me (i.e. from more central to not facial features): “This supports the idea that Chinese observers spend less time fixating facial features directly but differs from previous studies in that central fixations are not more common in Chinese observers.”

5) Just my opinion: Based on experiment 2 and the general discussion, I wonder if it is better to use a different word than culture for referring to Canadian vs. Chinese in the title. It is clear that environmental influences such as urbanization etc. cannot all be wrapped together as culture, and since cultural differences (especially east vs. west) usually automatically make people think about the collectivist/individualist axis, I think the massage of the paper may become clearer if the title was something like “Differences in Eye Movements for Face Recognition between Chinese and Canadians are not Modulated by Social Orientation”.

6) Abstract last sentence: I am not sure if the findings support a ‘more complex’ mechanism. I suggest changing to ‘other mechanisms than social orientation’ or something similar that makes a clearer conclusion.

7) I couldn’t access the data here osf.io/b5tdy and needed permission after logging into OSF

Minor:

1) Your reference numbers don’t seem to match between main text and references (e.g. Kelly et al is 34 and Liu et al is 33 in references but are 33 and 32 in the main text).

2) Page 7 near bottom: … that recommend a {minimum} sample size of …

6. PLOS authors have the option to publish the peer review history of their article (what does this mean?). If published, this will include your full peer review and any attached files.

Reviewer #1: **Yes: **Jiří Čeněk

Reviewer #2: No

Reviewer #3: No

---

## [Author Response · Author response to Decision Letter 0]

26 Sep 2023

Dear Dr. Coutrot,

We would like to thank you for considering our paper for publication in PLOS One. Enclosed with this letter are our answers to all comments on the manuscripts, including yours and those of the three reviewers. We are thankful to all involved for their comments, and believe they enrich our work and allow it to be clearer and more nuanced. We apologize for the delay in submitting these revisions, and are immensely grateful that you allowed us an extension. 

Editor

In particular, as you noted in your cover letter, the word "culture" should be used with care. You need to clearly state which cultural features are probed with your design.

We thank you for this comment. We follow Heine’s definition of culture, which is commonly used in cultural psychology research. We’ve added the following paragraph at the end of the introduction to clarify what we meant by culture:

Culture is a broad term that refers to a wide variety of concepts (ex. social norm, upbringing, belief systems, geographic location, etc.). Previous research has shown that country borders represent a relatively good proxy of cultural variations (44, 45). In line with this idea, in the present article we define groups of individuals coming from East-Asian and Western countries as “culturally different”, in the sense that they have grown up in environments with different sets of values, social norms, and so on. 

Also, Figure 5 is hard to interpret, you use bar graphs which hide the underlying distribution of the data, see https://garstats.wordpress.com/2016/03/09/one-simple-step-to-improve-statistical-inferences/

Please consider using an alternative such as violin plots, raincloud plots, or even simply add individual data points.

We thank you for this comment. We have replaced the figure with violin plots at your suggestion, which should help better interpret the data.

Reviewer #1: 

Comments:

1) I think it is unnecessary to repeat the same information for the both Experiments (mainly relevant for Material and Stimuli and Apparatus sections. This information might be mentioned before sections focused specifically on both experiments (and pointing out possible deviations of experimental conditions in E1 and 2). It would shorten the text of the paper and improve its readability.

We thank the reviewer for this comment. We have made sure to proofread the paper once more and remove as much redundancy as possible and improve readability throughout. That being said, while looking at the Material & Stimuli and Apparatus sections, most of the information here varies between experiments. As experiments 1 & 2 were not conducted in the same laboratories, the screens and eye-trackers used are different. As such, while we did our best to remove redundancy, these sections could not be shortened much further.

2) p. 7, participants: I would suggest reporting full G*Power settings including alpha. I was able to replicate the power analysis in G*Power but I was guessing with some settings.

We thank the reviewer for this suggestion. We have included more details in order to facilitate replicating the power analysis with G*Power for both experiments.

3) To be honest, I am not surprised that there was no main effect of priming found in study 1. Our own experiments on priming individualism always failed to cause any measurable differences in cognition to such an extent that I don't believe that simple semantic priming of this type can elicit any changes in cognition despite the results of metaanalysis conducted by Oyserman and Lee (2008). Recently, priming was criticized from left and right. it might be useful to reflect this debate (Sherman, 2021).

We thank the reviewer for their insights. We have added the following paragraph to discuss this issue in more detail.

“While a metanalysis by Oysermann & Lee(79) has shown that priming of collectivism and individualism successfully caused changes in cognition across many studies, priming paradigms have been criticized by many authors recently due to poor replicability. Different definitions of what constitutes priming “social” constructs(80) as well as differences in study design(36) can lead to different conclusions when evaluating the replicability of these studies. We believe that in-depth research on the mechanisms behind individualism/collectivism priming effects, as well as the replicability of priming studies, is necessary before further examining the question using priming paradigms.”

4) The lack of relationship between social orientation and facial recognition replicates the results of other cognitive style research using different stimuli such as maps, or Navon Figures (Lacko, 2020).

We thank the reviewer for this information! We’ve added a mention of these results in the discussion;

“...In Experiment 2, we also failed to reveal a clear association between social orientation questionnaires and eye movements. This is consistent with previous results by Lacko et al.(63),who showed that social orientation did not predict changes in processing of different stimuli such as maps or Navon letters.”

5) Moreover, I believe that the classic questionnaires that measure INDCOL/self-construals don’t hold their factor structure outside Western countries unless they are reduced to minimum number of items (Lacko, 2022) or extended to n-dimensions. I believe that despite the study conducted by Vignoles et al. (since it reports final model parameters far below generally accepted thresholds in MGCFA invariance research).

We thank the reviewer for their thoughts on the matter. We have added this passage to the discussion to discuss this.

“With respect to measures of social orientation, while the questionnaires used in Experiment 2 were extensively validated in prior studies, questionnaire research for cultural psychology as a whole has some serious limitations. While it has been widely used to study varied phenomena, many potential biases can affect validity (ex. construct inequivalence, item equivalence, response biases, etc.) As shown by Lacko et al.(67), analyzing while maximizing measurement invariance on a questionnaire can lead to radically different interpretations than simply analyzing the questionnaire in a “traditional” fashion. From their findings, concerns arise that factor structure may not hold when comparing full questionnaires between cultures. Even though studies have shown factor structure to be preserved(68), reported model parameters are much lower than average. In this study, we wished to use comparable measures as what was used before, so we opted for questionnaires. However, we recommend exploring different measures, perhaps more behavioral in nature, to ensure better comparability between cultures.”

6) I did not go through the validation studies of AICS and HVICS. Were they validated on Swiss and Chinese populations? If not was factor structure and reliability verified? From my own experience, horizontal-vertical INDCOL structure doesn’t always hold for example in postcommunist European countries.

Both questionnaires have been validated for use in Chinese populations, showing sufficient psychometric properties (HVICS - Li Y, Wang M, Wang C, Shi J. Individualism, collectivism, and Chinese adolescents' aggression: intracultural variations. Aggressive Behavior. 2010;36(3):187–94. pmid:20205262; AICS; Shulruf, B., Alesi, M., Ciochină, L., Faria, L., Hattie, J., Hong, F., ... & Watkins, D. (2011). Measuring collectivism and individualism in the third millennium. Social Behavior and Personality: an international journal, 39(2), 173-187.). The AICS and HVICS have also been validated with Swiss participants (Györkös, C., Becker, J., Massoudi, K., Antonietti, J. P., Pocnet, C., de Bruin, G. P., & Rossier, J. (2013). Comparing the horizontal and vertical individualism and collectivism scale and the Auckland individualism and collectivism scale in two cultures: Switzerland and South Africa. Cross-Cultural Research, 47(3), 310-331.), although these particular studies were done on the french version of the questionnaire.

7) Minor: You report some statistics, e.g. p-values without "0" (p < .05) while other, e.g. Cohens d with it. This could be made consistent throughout the text.

We thank the reviewer for this comment, we missed this while revising the paper. We have uniformized statistics to have a more consistent style throughout the paper. 

8) p. 31, suggestion to choose a wider variety of questionnaires: I am not entirely sure that this will solve the problem, especially in comparisons of multiple cultural samples. The first obstacle you will encounter will be the validation of the questionnaires in countries beyond the traditional USA/Western Europe vs. China/Japan/Korea samples. Then the second obstacle will be that many of the possible cross cultural differences disappear once you aggreggate the culture means. I believe that a promising approach might be to more finely identify participant profiles within each country/culture and compare countries based on the proportion of those profiles within country level samples (Na, 2020). Another promising way is to design more behavioral oriented measures, something in line with research conducted by Talhelm. This is my personal opinion based on the problems we encountered in our research that are very similar to the current study.

We thank the reviewer for these very insightful suggestions! We agree with your points, and had not thought about the colossal cross-cultural validation studies this would imply. We have removed this suggestion. 

We also agree with you that more behavioral-oriented measures are promising. We have added this passage to the discussion to discuss this:

“However, we recommend exploring different measures for future studies, perhaps more behavioral in nature, to ensure better comparability between cultures. Research by Talhlem and collaborators(69–71) observe participants’ behavior in controlled contexts to make inferences, instead of asking questions via questionnaire. While this can complexify data collection, it remains a promising avenue for cultural research.”

Reviewer #2:

Major points

- The social orientation questionnaires resulted in unexpected patterns, which is not an issue in itself of course. However, it brings up questions about the nature of these unexpected findings (e.g., is it due to the validity of measures, or the selected participants, or is the notion of individualism/collectivism simply ‘archaic’ as the authors also suggest, etc.). The unexpected findings and the closer examination of subscales in the context of culture give rise to a discussion in itself. This makes any conclusions regarding links with cultural differences in face fixations very difficult. It seems that a manuscript focusing on measuring social orientation and deconstructing findings inconsistent with the current literature is useful before jumping to examine links with cultural differences in face viewing.

We thank the reviewer for their insights and wholeheartedly agree that more research on the measurement of social orientation is needed before continuing to try and explain cultural differences. We have added the following passage in the introduction to specify why we chose to proceed with priming and questionnaires:

“We conducted two experiments to test if both primed and self-reported social orientation modulate eye movements during an Old/New face recognition task(13,35). To be consistent with previous social orientation studies, we have chosen designs that are similar to previous studies, both in the methods and questionnaires used. In Experiment 1, we probed the existence of a link between individualism/collectivism (IND/COL) and eye movements during face recognition by implementing an IND/COL priming paradigm. We used a within-subjects design as they have been shown to have better statistical power in priming paradigms(36). This method is similar to the one used in Liu et al.(33) However, unlike the latter study, we included both West European and Chinese participants.”

We have also added this paragraph to the discussion to add nuance to our interpretations.

“However, due to the many unexpected results that were found in our study, we should be careful in our interpretations. The fact that the eye-movement patterns we found were inconsistent with previous studies, and that the priming paradigm may not have been successful in causing differences to emerge make the interpretation of our results difficult. More research is needed to fully understand cultural expressions of individualism and collectivism across different countries before studying this topic further. That being said, we did not manage to reveal an effect of social orientation on eye movements with methods that were consistent with what was previously used in social orientation research. Further replications using more controlled experimental parameters, measures of success in the priming task and more behavioral measures of social orientation would benefit immensely and give valuable insight to the field.”

- Exp1: why were participants primed in a within-subjects design? It would seem surprising to change eye movements so quickly after each priming condition.

We thank the reviewer for this comment. We chose to use a within-subject design as it has been shown to have more statistical power to detect priming effects (Rivers & Sherman, 2019). Rivers & Sherman show that using a within-subject design allows them to replicate four out of four of their selected studies, while between-subject designs allowed only one successful replication. We have added the following passage at the end of the introduction to explain our reasoning:

“In Experiment 1, we probed the existence of a link between individualism/collectivism (IND/COL) and eye movements during face recognition by implementing an IND/COL priming paradigm. We used a within-subjects design as they have been shown to have better statistical power in priming paradigms(36).”

However, we agree that for effects on low-level cognitive processes such as eye-movements, a within-subject design with a very short delay between priming and measurement may not have been able to detect any changes. As such, we have nuanced our interpretation in the discussion, like so:

“Finally, there is a possibility that the priming task used in Experiment 1 was not effective. We had no way of knowing whether the priming was effective on participants, and it is possible that a within-subject design was not appropriate to detect differences in eye-movement patterns. However, it is crucial to note that the pronoun circling manipulation used here has previously lead to many positive results with a variety of tasks(32,33,77,78). In addition, the current study replicates the absence of a social orientation priming effect on eye movements reported in Liu et al.(33), using the same priming manipulation and face recognition task, but using a between-participant design.”

- exp1: we have no way of knowing whether priming was (subjectively) achieved or not. In light of the lack of priming effects, we cannot conclude whether priming was not achieved, of priming was achieved but did not modulate eye movements. Priming did not elicit expected differential fixational viewing, and the hypothesised cultural differences in face scanning patterns were not observed. Yet, the authors conclude that ‘priming individualistic & collectivistic social values did not seem to abolish the cultural differences in eye movements for face recognition’. First, it is unclear whether these social values were successfully primed. Secondly, the cultural differences in question manifest in an unexpected pattern and any conclusion should therefore be more carefully worded.

We thank the reviewer for this comment. We have modified the discussion of experiment 1 to add more nuance and mention these points.

“In experiment 1, we revealed significant cultural differences that were atypical compared to previous results, and priming did not seem to have any effect on participants. It is unclear whether this is because there is no effect of priming or because the priming procedure simply failed. Our data show a qualitative cultural contrast, with more fixations towards the eyes for West Europeans and more central fixations for East Asians. Analyses with iMap4 yielded no significant results. However, subsequent ROI analyses did reveal greater fixation distribution around the face for Chinese participants compared to Swiss participants. This supports the idea that Chinese observers spend more time than Swiss observers fixating areas peripheral to the main features. However, it differs from previous studies in that Chinese and Swiss participants did not differ in the time spent fixating the main facial features nor the center of the face.. This difference may be due to the fact that our Chinese sample was recruited from a different region than the one from Blais et al.13. This can still be seen as consistent with the idea proposed by Caldara et al.14 and Miellet et al.15 that East-Asians process facial features in peripheral vision. Further investigation would be necessary to examine how samples collected from different regions within the same country may show different patterns of visual strategies.”

- While some cultural differences in face scanning were observed, they were not always as expected/consistent with the literature (e.g., exp2). Again, while this is not an issue in itself, the manuscript needs to be clearer that a differential pattern was observed and discuss inconsistencies with previous findings.

We thank the reviewer for this comment. We made sure to add clarifications throughout the manuscript in order for readers to easily understand how our results differ from previous findings.

- How was data quality of eye tracking data assessed (beyond drift correction, e.g., % data loss, precision, etc. during trials)? This is important to establish that the greater distribution in the Chinese group was not the result of poorer data quality. This is particularly relevant given that the faces were presented in the corner of the screen, i.e., at locations where eye tracking quality tends to drop off.

We thank the reviewer for their comment. Following your recommendation, we assessed data quality in two ways: 1) By measuring the percentage of data loss in each sample; 2) By comparing the measure of error during the calibration for both samples. With regard to data loss, we’ve measured the mean percentage of unusable samples due to blinks or failing to detect the participants’ eyes. Samples of eye movements were measured every 4ms. We sadly do not have access to this measure for experiment 1, due to differences in the eye tracking equipment used between experiments. For experiment 2, we found that the Chinese sample indeed had a significantly higher percentage of data loss (6.74%) compared to the Canadian sample (2.23%). The second measure came from a validation procedure, using Eyelink software, that follows calibration and allows measuring its quality. A higher measure of error means that actual gaze position at the calibration points has higher possible deviation. Computing a two sample t-test revealed that there was no significant difference in error between the two samples (t(97) = 1.842, p = .069, Cohen’s d = .370). The mean error was actually higher, although not statistically significant, in the Canadian sample (.541 degrees) than in the Chinese sample (.442 degrees). We have added the following to the results section of Experiment 2:

In order to evaluate whether this difference was due to poorer data quality, we measured the mean percentage of data loss per trial for every participant in both the Canadian and Chinese samples. A paired t-test revealed that there was a significantly higher percentage of data loss in the Chinese sample (t(97) = 5.235, p <.001, Cohen’s d = 1.052). However, there is no significant difference in error after calibration between the two samples (t(97) = 1.842, p = .069, Cohen’s d = .370), with error being slightly higher on average in the Canadian sample (.542 degrees) compared to the Chinese sample (.442 degrees). As such, there is little chance our results are caused by poor data quality, even though higher data loss in the Chinese sample is regrettable. These analyses could not be run for experiment 1, due to the format data was saved in not including measures of data quality.

The following was also added to a paragraph of the general discussion:

…However, instead of displaying the usual central fixation bias observed with East Asian populations, our Chinese participants instead fixated more significantly around the face, but not on facial features. This finding is inconsistent with previous studies. Analyses on data quality suggest this is not due to poorer data quality in the Chinese sample, at least for experiment 2. This behavior could qualitatively and conceptually remain in line with common East Asian holistic/global attentional patterns, and is consistent with strategies employed during scene perception(30). That being said, more research is needed to see if this pattern can be replicated. 

Minor points

- “here we included cross-cultural participants of West European and Chinese descent.” Rephrase: ‘included both West European and Chinese participants’, rather than ‘descent’ which has wider meaning.

We thank the reviewer for this suggestion. We rephrased this sentence as suggested.

Reviewer #3: 

1) Intro: Page 4 lines 4-7: I suggest also mentioning evidence that does not support the value systems hypothesis, e.g. Knox, P. C. & Wolohan, F. D. Cultural diversity and saccade similarities: Culture does not explain saccade latency differences between Chinese and Caucasian participants. PloS one 9, e94424 (2014).; Kardan, O., Shneidman, L., Krogh-Jespersen, S., Gaskins, S., Berman, M. G., & Woodward, A. (2017). Cultural and developmental influences on overt visual attention to videos. Scientific reports, 7(1), 1-16.

We thank the reviewer for this suggestion, and for recommending this very interesting work. We’ve added a citation of both articles at the requested point in the article. We have also added the following paragraph to the introduction to address this work:

Regarding eye movement patterns, there have also been studies showing evidence that culture may not be what drives the observed differences between groups. Knox & Wolohan(23) have compared Chinese individuals raised in England to both English and Chinese participants, and have shown that while they are closer to British participants in their values, their eye movement patterns are identical to the Chinese group. Another study by Kardan et al. (24) comparing Mayan to US participants has shown that participants from the US fixate more on the background information during videos compared to Mayan children, which is not what would be expected with the social orientation hypothesis. While the authors do advise caution in applying the social orientation theory to their findings (it is unsure whether the individualism-collectivism continuum applies to Mayan culture), it still shows that not all of the influences attributed to culture are as clear-cut as they were expected to be.

2) Power: In general it is not recommended to estimate effect size from previous literature that had small sample size, because those effect sizes would be very likely overestimated. For example see Button, K. S., Ioannidis, J. P., Mokrysz, C., Nosek, B. A., Flint, J., Robinson, E. S., & Munafò, M. R. (2013). Power failure: why small sample size undermines the reliability of neuroscience. Nature reviews neuroscience, 14(5), 365-376.

See the next point.

3) In experiment 1, the power analysis assumes equal group sizes, so the N = 15 (which is fewer than 34/2 = 17) for Swiss is below the recommended minimum sample size.

We thank the reviewer for both precisions on power analyses. We apologize for forgetting to mention the unequal group sizes in the limitations section, and we agree that it should be mentioned that we may have overestimated effect sizes in the sample size calculation, particularly for experiment 1. We’ve added the following paragraph to the limitations section to address this: 

“First, some limitations concerning statistical power in our experiments need to be considered. While we calculated minimum sample size by using effect sizes reported in previous studies, these studies all had a relatively low number of participants. This is not recommended, as effect sizes detected on smaller samples tend to be overestimated(65). Also, in experiment 1, we computed a minimum sample size of 34 participants. However, this assumes equal sample sizes between groups, yet we only have 15 Swiss participants. This is below the recommended sample size and may prevent us from detecting the effect. Taken together, it is a possibility that our study may have been underpowered, making it more difficult to reveal an effect. That being said, we believe that experiment 2 alleviates this limitation as it has a bigger sample size and as such should not be underpowered.”

That being said, we believe that experiment 2 alleviates this limitation as it has a bigger sample size and as such should not be underpowered. Following your comments, we had internal discussions about removing experiment 1 from the paper due to its many flaws. However, due to the fact that our study is not the only one that failed to reveal priming effects of individualism/collectivism on visual perception, and to avoid the “file-drawer” problem, we chose to leave the experiment in the paper. While flawed, we believe there is still meaningful information to be drawn from this experiment.

4) In experiment 1, isn’t the finding that Chinese participants fixated on ‘rest of face’ more than Swiss in the opposite of expected direction? This sentence in page 15 seems like moving the goalposts a bit to me (i.e. from more central to not facial features): “This supports the idea that Chinese observers spend less time fixating facial features directly but differs from previous studies in that central fixations are not more common in Chinese observers.”

We recognize our wording could have given this impression, but we disagree that we are “moving the goalpost.” We believe our results are still consistent with previous studies. The proposed explanation for why East Asian participants make more central fixations than Westerners is that they use peripheral vision to process facial features, as opposed to central vision in Westerners. This has been backed by Caldara et al. (14) as well as Miellet et al. (15), Tardif et al. (10) and Estéphan et al. (16). As such, even though fixations are not biased towards the center in our study (which we agree was a surprising finding), their fixations are mostly outside of main facial features. Consequently, it is reasonable to assume that they would also use peripheral vision to process facial features. 

As for why we did not find the same pattern, while both our study and Blais et al. (2008)’s study compared Westerners to Chinese and Japanese participants, the Chinese participants were not recruited from the same region of China, and may not allow previous results to generalize as well as they could. I believe this could be explored further in order to quantify how differences in upbringing even within the same culture could affect visual perception.

We added the following to the manuscript to explain our thoughts. We hope that this adequately answers your questions.

“However, subsequent ROI analyses did reveal greater fixation distribution around the face for Chinese participants compared to Swiss participants. This supports the idea that Chinese observers spend more time than Swiss observers fixating areas peripheral to the main features. However, it differs from previous studies in that Chinese and Swiss participants did not differ in the time spent fixating the main facial features nor the center of the face. This difference may be due to the fact that our Chinese sample was recruited from a different region than the one from Blais et al.13. This is still consistent with the idea proposed by Caldara et al.14, Miellet et al.15, Tardif et al. (10) and Estéphan et al. (16) that East-Asians process facial features in peripheral vision. Further investigation would be necessary to examine how samples collected from different regions within the same country may show different patterns of visual strategies.”

5) Just my opinion: Based on experiment 2 and the general discussion, I wonder if it is better to use a different word than culture for referring to Canadian vs. Chinese in the title. It is clear that environmental influences such as urbanization etc. cannot all be wrapped together as culture, and since cultural differences (especially east vs. west) usually automatically make people think about the collectivist/individualist axis, I think the message of the paper may become clearer if the title was something like “Differences in Eye Movements for Face Recognition between Chinese and Canadians are not Modulated by Social Orientation”.

We thank the reviewer for this comment, and agree this could be confusing. We had initially used the word culture as it is frequently used in these types of studies. While we think it is still relevant in the text itself, for the selfsame reasons you gave, we agree that changing it in the title can help make the paper’s objective clearer. As such, we changed the title to the following, as per your suggestion: 

“Differences in Eye Movements for Face Recognition between Canadian and Chinese participants are not Modulated by Social Orientation”

We have also more clearly defined the meaning “culture” in this work in the introduction, as seen in this paragraph added to the introduction: 

Culture is a broad term that refers to a wide variety of concepts (ex. social norm, upbringing, belief systems, geographic location, etc.). Previous research has shown that country borders represent a relatively good proxy of cultural variations (44, 45). In line with this idea, in the present article we define groups of individuals coming from East-Asian and Western countries as “culturally different”, in the sense that they have grown up in environments with different sets of values, social norms, and so on. 

6) Abstract last sentence: I am not sure if the findings support a ‘more complex’ mechanism. I suggest changing to ‘other mechanisms than social orientation’ or something similar that makes a clearer conclusion.

We thank the reviewer for this comment. We agree that reformulating would be best. We’ve changed the last sentence of the abstract to the following:

“Cultural differences in eye movements for faces might originate from mechanisms distinct from social orientation.” 

7) I couldn’t access the data here osf.io/b5tdy and needed permission after logging into OSF

Thank you for reporting this. This should now be fixed; Use the following link to access the project: 

https://osf.io/b5tdy/?view_only=c4c315f2249044e1a6964ab2118153f0

Minor:

1) Your reference numbers don’t seem to match between main text and references (e.g. Kelly et al is 34 and Liu et al is 33 in references but are 33 and 32 in the main text).

We thank the reviewer for pointing out this issue. This has now been fixed in the revised version of the manuscript.

2) Page 7 near bottom: … that recommend a {minimum} sample size of …

We thank the reviewer for this precision. We rephrased this.

We remain available should you need to further discuss these changes, or require further changes be made to the manuscript. We hope you will favorably consider publication of this work in your journal.

Sincerely yours,

Francis Gingras, Amanda Estéphan, Daniel Fiset, He Lingnan, Roberto Caldara, Caroline Blais

---

## [Decision Letter · Decision Letter 1]

20 Nov 2023

Differences in Eye Movements for Face Recognition between Canadian and Chinese participants are not Modulated by Social Orientation

PONE-D-23-11102R1

Dear Dr. Blais,

We’re pleased to inform you that your manuscript has been judged scientifically suitable for publication and will be formally accepted for publication once it meets all outstanding technical requirements.

Please also address R2's last minor comments.

Kind regards,

Antoine Coutrot

Academic Editor

PLOS ONE

Additional Editor Comments (optional):

Reviewers' comments:

Reviewer's Responses to Questions

**Comments to the Author**

1. If the authors have adequately addressed your comments raised in a previous round of review and you feel that this manuscript is now acceptable for publication, you may indicate that here to bypass the “Comments to the Author” section, enter your conflict of interest statement in the “Confidential to Editor” section, and submit your "Accept" recommendation.

Reviewer #1: All comments have been addressed

Reviewer #2: (No Response)

Reviewer #3: All comments have been addressed

2. Is the manuscript technically sound, and do the data support the conclusions?

Reviewer #1: Yes

Reviewer #2: Yes

Reviewer #3: (No Response)

3. Has the statistical analysis been performed appropriately and rigorously? 

Reviewer #1: Yes

Reviewer #2: Yes

Reviewer #3: (No Response)

4. Have the authors made all data underlying the findings in their manuscript fully available?

Reviewer #1: Yes

Reviewer #2: Yes

Reviewer #3: (No Response)

5. Is the manuscript presented in an intelligible fashion and written in standard English?

Reviewer #1: Yes

Reviewer #2: Yes

Reviewer #3: (No Response)

6. Review Comments to the Author

Reviewer #1: I reviewed the comments made by the authors since the last round of reviews. I believe the athors successfully addresed the comments made by me and the other reviewers.

Reviewer #2: The authors significantly changed the manuscript to incorporate and account for inconsistencies in findings compared to previous literature. The revised manuscript is clearer. I have a few remaining comments.

- Lines 114-117: suggestion that culture may not be driving eye movements. The example used for British Born Chinese with cultural values closer to British, but eye movements being more consistent with Chinese participants: this does not preclude the effect of cultural influence (e.g., identifying with more-British values, but implicit cultural learning throughout development via Chinese caregivers). For the next example suggesting US Americans focused more on background values compared to Mayan individuals and therefore showing inconsistent patterns with literature: the typical Western vs Eastern culture comparisons are relative, i.e., Westerners look less at the background compared to Easterners. Mayan individuals are a different group, and Westerners may be looking more at the background compared to Mayans (but possibly not Easterners).

- Line 498: t-test values for individualism and collectivism are the same – double check.

- Line 117: replace “identical”. E.g., ‘consistent with’ or ‘in line with’ etc.

- Line 121: replace “unsure” with “unclear” or “unknown” etc.

Reviewer #3: (No Response)

7. PLOS authors have the option to publish the peer review history of their article (what does this mean?). If published, this will include your full peer review and any attached files.

Reviewer #1: **Yes: **Jiří Čeněk

Reviewer #2: No

Reviewer #3: No

---

## [Editor Report · Acceptance letter]

5 Dec 2023

PONE-D-23-11102R1 

Differences in Eye Movements for Face Recognition between Canadian and Chinese participants are not Modulated by Social Orientation 

Dear Dr. Blais:

I'm pleased to inform you that your manuscript has been deemed suitable for publication in PLOS ONE. Congratulations! Your manuscript is now with our production department. 

Kind regards, 

on behalf of

Dr. Antoine Coutrot 

Academic Editor

PLOS ONE